

# Characterisation of aerosol provenance from the fractional solubility of Fe (Al, Ti, Mn, Co, Ni, Cu, Zn, Cd and Pb) in North Atlantic aerosols (GEOTRACES cruises GA01 and GA03) using a two stage leach

Rachel U. Shelley[1, 2, 3], William M. Landing[1], Simon J. Ussher[2], Helene Planquette[3], and Geraldine Sarthou[3]

[1]Dept. Earth, Ocean and Atmospheric Science, Florida State University, 117 N Woodward Ave, Tallahassee, Florida, 32301, USA
[2]School of Geography, Earth and Environmental Sciences, University of Plymouth, Drake Circus, Plymouth, PL4 8AA, UK
[3]Laboratoire des Sciences de l'Environnement Marin, UMR 6539 LEMAR (CNRS/UBO/IRD/IFREMER), Institut Universitaire Européen de la Mer, Technopôle Brest-Iroise, Plouzané 29280, France

*Correspondence to*: Rachel U. Shelley (rshelley@fsu.edu)

**Abstract.** The fractional solubility of aerosol-derived trace elements deposited to the ocean surface is a key parameter of many marine biogeochemical models. Yet, it is currently poorly constrained, in part due to the complex interplay between the various processes that govern the solubilisation of aerosol trace elements. In this study, we used a sequential two-stage leach to investigate the fractional solubility of a suite of aerosol trace elements (Al, Ti, Fe, Mn, Co, Ni, Cu, Zn, Cd and Pb) from samples collected during three GEOTRACES cruises to the North Atlantic Ocean. Regardless of the leaching protocol used (mild versus strong leach), the same trends were observed. These were that trace elements from aerosols from 1) North Africa were always the least soluble, and the most homogeneous (e.g. Fe was $0.36 \pm 0.12$ % and $6.0 \pm 1.0$ % soluble in North African and $6.5 \pm 5.5$ %  and $17 \pm 11$ % soluble in non-African aerosols following leaches with ultra-high purity water, and 25 % acetic acid, respectively), 2) aerosols from the most remote locations were generally the most soluble, but had the most spread in the values of fractional solubility and 3) primarily pollution-derived TEs (Ni, Cu, Zn, Cd and Pb) were significantly enriched above crustal values in aerosols, even in samples of North African origin.  We present aerosol trace element solubility data from two sequential leaches that provides a "solubility window", covering a conservative, lower limit to an upper limit, the maximum potentially soluble fraction, and demonstrate why this lower limit of solubility may underestimate aerosol TE solubility in some regions. The leaching technique that yields the upper limit can also be used to estimate trace element solubility from suspended particulate matter (SPM). Therefore, facilitating direct comparison with SPM leached using the same technique, thereby introducing some degree of standardisation between aerosol and SPM trace element solubility studies which may help inform of in-water processes that modify the solubility, and thus bioavailability, of atmospheric particles following deposition to the surface ocean.




## 1. Introduction

Aerosol trace element (TE) solubility is a key parameter of many biogeochemical models, but it is poorly constrained, e.g. Fe solubility estimates range from 0.001-90% (Aguilar-Islas et al., 2010; Baker et al., 2016). The fractional solubility (herein referred to as "solubility") of aerosol TEs is defined in terms of the amount of a TE in solution from any given leach that passes through a filter (usually < 0.45 or 0.2 µm), expressed as a percentage (Baker and Croot, 2010; Baker et al., 2016; Jickells et al., 2016). This operational definition accounts for some of the variability in published values. A number of factors impact aerosol TE solubility, such as: (1) the choice of leaching protocol, and (2) the aerosol provenance, which in turn is impacted by a combination of factors such as the mineralogy of the particles, acid processing during atmospheric transport, and the presence/absence of emissions resulting from e.g. vehicles, industry and agricultural practices. Several studies have concluded that the most significant effects on aerosol Fe solubility result from the source/composition of the aerosols, rather than changes in physico-chemical parameters, such as temperature, pH and oxygen concentration of the leach medium, or the choice of batch versus flow-through techniques (e.g. Aguilar-Islas et al., 2010; Fishwick et al., 2014).

There have been a number of studies that have focused on the role of aerosol TEs on biogeochemical cycles in the North Atlantic (e.g. Sarthou et al., 2003; Baker et al., 2013; Buck et al., 2010; Ussher et al., 2013; Powell et al., 2015). More recently, the GEOTRACES programme has produced a number of aerosol datasets, which has stimulated further discussion on the use of this data to look for trends that link TE solubility and aerosol provenance (e.g. Baker et al., 2016; Jickells et al., 2016). Elemental ratios, enrichment factors and air mass back trajectory simulations have long been used as a first approximation of aerosol source, and there are many studies that employ multivariate statistical analyses for aerosol source apportionment (e.g. Chueinta et al., 2000; Laing et al., 2015). In addition, more studies are making use of stable isotope ratios to investigate aerosol provenance. Some of these methods are well-established and have a relatively long history of use in this purpose, such as Pb isotopes (e.g. Maring et al., 1987), and Sr and Nd isotopes (e.g. Skonieczny et al., 2011; Scheuvens et al., 2013 and references therein), and data from investigations of novel isotope systems are increasing. For example, Fe isotopes show promise as a way to differentiate between anthropogenic and mineral dust aerosols (Conway et al., submitted). In contrast, Cd isotopes may not be a suitable tool for aerosol source apportionment (Bridgestock et al., 2017).

The leachable (soluble) fraction of aerosol TEs is used as a first approximation of the bioavailable fraction. Therefore, experimental conditions should mimic natural conditions as closely as possible, while yielding reproducible results. Ideally, the leach protocol used fits both these criteria. However, that is not





always strictly possible for reasons such as access to the leach medium of choice, availability of analytical
instrumentation, and cost. Currently, however, there is no standardised aerosol leaching protocol, but it is
recognised that this should be a priority for future studies (Baker et al., 2016). Some commonly-used
leach media are ultra-high purity (UHP) water (18.2 M$\Omega$.cm), seawater, weak acids (e.g. 1% HCl, 25 %
acetic acid), or ammonium acetate buffer (e.g. Buck et al., 2006, Baker et al., 2006; Berger et al., 2008).
Given that UHP water and rain water have broadly similar pH (~ pH 5.6), UHP water is used as an
analogue for rain/wet deposition, as wet deposition is thought to dominate the supply of many TEs, at
least at some regional and local scales (Helmers and Shremms, 1995; Kim et al., 1999; Powell et al.,
2015). However, the solubilities estimated from the UHP water "instantaneous" leach (Buck et al., 2006),
a flow-through method where the leach medium is in contact with the aerosols for 10 - 30 s, may be
higher than those resulting from the seawater "instantaneous" leach, due to the extremely low ionic
strength of UHP water. As such, freshly-collected, filtered seawater may yield more environmentally-
relevant data, but can be more challenging to analyse, although analytical capabilities are rapidly
improving. Nevertheless, as solubility for many TEs has been shown to be of a second order type (initial
fast release, followed by a slower sustained release with time; e.g. Kocak et al., 2007; Mackey et al.,
2015), the instantaneous leach likely yields conservative lower limit estimates of TE solubility due to the
short contact time between the aerosols and leach medium.
In order to estimate the upper limit of TE solubility, and provide a "solubility window", a more aggressive
leach is required. In this study, we have taken this approach, and estimate an upper limit of TE solubility
using a leach protocol more commonly used to estimate TE solubility from suspended particulate matter
(SPM; Berger at al., 2008). Therefore, a two-step, sequential leach approach was employed: (i)
instantaneous UHP water leach to mimic the initial rapid release of TEs into rain drops and the surface
mixed layer of the ocean, and (ii) 25 % acetic acid leach to mimic the slower and sustained release from
aerosol particles during the residence time in the euphotic zone. In addition to the aerosol TE solubility
data, soluble major anion ($NO_3^-$ and non-sea salt (nss-) $SO_4^{2-}$) data are also discussed.
The instantaneous leach can be conducted using UHP water or sewater as the leach medium. The
advantages of conducting it using UHP water are that UHP water is a reproducible medium (allowing for
inter-lab comparisons), which can be easily analysed by ICP-MS for many elements simultaneously
without the need for time-consuming sample handling steps such as separation techniques, and drying
down and re-dissolving the residue. The UHP water leaches can easily be conducted at sea, or in the home
laboratory. In addition, aliquots can be taken to determine acid species (e.g. $NO_3^-$, $SO_4^{2-}$), which can be





used to estimate aerosol acidity; an important control of aerosol metal solubility (e.g. Meskhidze et al.,
2003; Solmon et al., 2009; Paris et al., 2011).
While UHP water can be thought of as an analogue for rain water (i.e. wet deposition), the extremely low
ionic strength of UHP water, and the absence of the metal binding ligands naturally present in rain water
and sea water (e.g. Chieze et al., 2012; Wozniak et al., 2014), means that UHP water is not a perfect
analogue for oceanic receiving waters. However, freshly-collected, filtered (< 0.2 µm) sea water may be
substituted for UHP water. It is assumed that the use of such water would likely produce a better estimate
of the fractional solubility of TEs on first contact with the receiving waters. Despite this, sea water is a
complex matrix which presents some analytical challenges, and the leaching experiments must be
conducted in the field if fresh sea water is to be used. For Fe, leaches using UHP water (~ pH 5.6)
typically produce higher solubility estimates than leaches conducted with natural seawater (~ pH 8.2) due
to the pH sensitivity of dissolution and the higher ionic strength of sea water. On occasions where higher
solubility in seawater is observed, complexation by Fe binding ligands is likely the cause.  However, the
short contact time between the aerosols and leaching solution during the instantaneous leach results in a
conservative, lower limit of solubility, regardless of the choice of leaching medium.
In contrast, the 25 % acetic acid leach provides an upper-limit of what is potentially soluble over the life
time of aerosol particles in the euphotic zone. The pH of this leach (2.1) is just below those of
zooplankton or fish digestive tracts and the reducing agent mimics the low oxygen environments inside
faecal pellets and marine snow aggregates. Indeed, Schmidt et al. (2016) have demonstrated that
lithogenic Fe is mobilised in the gut passage of krill resulting in threefold higher Fe content in the muscle,
and fivefold higher Fe content of the faecal pellets of specimens close to lithogenic source material
compared to those from offshore. Use of this technique also allows direct comparison of aerosol and
marine particle solubility data, which can be useful when investigating SPM provenance (e.g. terrestrial
versus biogenic).
To investigate the regional variation in the solubility of key TEs in North Atlantic aerosols using the two-
stage leach, samples were collected during the US-GEOTRACES GA03 campaigns in 2010 and 2011,
and the French GEOTRACES GA01 campaign in 2014 (www.geotraces.org). Both campaigns took place
in the North Atlantic Ocean, with GA03-2010 and GA01 departing from Lisbon, Portugal.  The cruise
tracks were designed to traverse a wide variety of biogeochemical provinces (Longhurst, 2010), from
continental shelf regions, to an eastern boundary current upwelling system (off West Africa), the
oligotrophic North Atlantic gyre, and sub-Arctic waters (North Atlantic Deep Water formation region),
and span a large gradient in atmospheric dust loading. The focus of this paper is Fe and the GEOTRACES

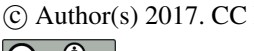


reference elements, Al, Cd, Cu, Mn, Pb, Zn, plus Co, Ni, and Ti. This suite of TEs includes bioactive
elements, tracers of atmospheric deposition, and pollutants. Some TEs fit into more than one of these
categories. Here, we use the term 'trace element' in the context of open ocean water column
concentrations, thus acknowledging that elements such as Al, Fe and Ti are not present in trace
abundances in aerosol source material. Aerosol concentrations for a suite of other elements (Li, Na, Mg,
P, Sc, V, As, Se, Rb, Sr, Sn, Sb, Cs, Ba, La, Ce, Nd, Th, U) were also determined, but will not be
discussed further here. However, these data are available at BCO-DMO (GA03; www.bco-dmo.org/) and
LEFE-CYBER (GA01; (www.obsvlfr.fr/proof/php/GEOVIDE/GEOVIDE.php), and on request from the
lead author.

## 2. Methods

### 2.1. Aerosol sample collection

Aerosol samples (n=57) were collected in the North Atlantic Ocean aboard the *R/V Knorr* during the *US-*
*GEOTRACES GA03* cruises (15 Oct – 2 Nov 2010 and 6 Nov – 9 Dec 2011, and aboard the *N/O Pourquoi*
*Pas?* during the French GEOTRACES GA01 cruise (GEOVIDE, 15 May – 30 June 2014) (Fig. 1). The
aerosol collections have been described previously (Wozniak et al., 2013; 2014; Shelley et al., 2015;
2017). Briefly, air was simultaneously pulled through twelve acid-washed 47 mm diameter Whatman 41
(W41) ashless filter discs at approximately 1.2 m$^3$ min$^{-1}$ (134 cm s$^{-1}$ face velocity) using a high-volume
aerosol sampler (model 5170-VBL, Tisch Environmental). The metadata and concentration data for the
aerosol leaches can be found in the supplementary information (Table S1). All filters were stored frozen (-
20°C) and double bagged prior to processing, both on the ship and upon returning to the home
laboratories.
To avoid contamination from the ship's stack exhaust, aerosol sampling was controlled with respect to
wind sector and wind speed using an anemometer interfaced with a datalogger (CR800, Campbell
Scientific). The samplers were programmed to run when the wind was ± 60° from the bow of the ship and
> 0.5 m s$^{-1}$. When the wind failed to meet these two criteria, the motors were shut off automatically and
not allowed to restart until the wind met both the speed and direction criteria for 5 continuous minutes. In
addition, the samplers were deployed on the ship's flying bridge as high off the water as possible (~14 m
above sea level) to minimise collection of sea spray.

### 2.2. Trace element determination – totals aerosol TEs

For the determination of total aerosol TE loadings (Al, Ti, Mn, Fe, Co, Ni, Cu, Zn, Cd, Pb) the W41 filter
discs were digested in tightly-capped 15 mL Teflon-PFA vials (Savillex). Firstly, 1000 µL of ultrahigh
purity (UHP) 15.8 M nitric acid (Optima or Merck ultrapur) was added to each vial, heated to 150°C on a



hotplate, and then taken to dryness. Secondly, 500 µL of 15.8 M nitric acid (13.2 M HN0$_3$) and 100 µL of
28.9 M hydrofluoric acid (5.8 M HF) (Optima or Merck ultrapur) was added to each vial, re-heated to
150°C on a hotplate, then taken to near dryness. After the final digestion and evaporation step, the
samples were re-dissolved in 20 mL of 0.32 M nitric acid for analysis (Morton et al., 2013). All filter
digestions were performed under Class-100 laminar flow conditions. Total aerosol TE concentrations
were determined by magnetic sector field inductively coupled plasma mass spectrometry (ICP-MS;
Thermo Element-2) at the National High Magnetic Field Laboratory (NHMFL) at Florida State University
(FSU; GA03) or Pôle de Spectrométrie Océan (PSO) at the Institut Universitaire Européen de la Mer
(IUEM; GA01), France. Samples were introduced to a PFA-ST nebuliser (Elemental Scientific
Incorporated) via a modified SC-Fast introduction system consisting of an SC-2 autosampler, a six-port
valve and a vacuum rinsing pump. Replicate blank solutions for the acid digestions were prepared by
digesting W41 discs that had been deployed in the aerosol samplers for 1 h while not in operation, and the
resulting concentrations were subtracted from all acid-digested filter samples. Details of the blanks and
analytical figures of merit, including CRM recoveries, have previously been reported (Shelley et al., 2015;

180    2017).


**2.3. Trace element determination – soluble aerosol TEs**
In this study, we discuss the results from (1) an 'instantaneous' leach (Buck et al., 2006), that provides a
lower limit estimate of the most labile TE fraction (analogous to the fraction that dissolves immediately
on contact with water), followed by (2) a more protracted leach using 25 % acetic acid (with the reducing
agent, hydroxylamine hydrochloride, and heat, 10 min at 90 °C). As this second leach aims to access a
less labile fraction of the TEs of interest, without significantly attacking TEs bound within the mineral
matrix (Koçak et al., 2007; Berger et al., 2008), it may provide an upper limit estimate for the fractional
solubility of these aerosol TEs as the aerosols mix down into the ocean.
The "instantaneous" leach is a flow-through method using UHP water, conducted under a Class-100
laminar flow hood. Using this technique, 100 mL of UHP water (> 18 MΩ.cm resistivity, pH ~5.5,
Barnstead Nanopure) is rapidly passed through an aerosol-laden W41 filter held in a polysulfone vacuum
filtration assembly (Nalgene). Operationally-defined dissolved (≤ 0.45 µm) TEs are collected in the
filtrate (leachate) by positioning a GN-6 Metricel backing filter (cellulose esters) below the W41 disc in
the filtration assembly (Buck et al., 2006). In this study, the leachate was transferred to an acid-clean low
density polyethylene (LDPE) bottle and acidified to 0.024 M (~ pH 1.7) with UHP HCl and double-
bagged for storage until analysis at FSU or IUEM. As for total elemental determinations, soluble TEs in





the leachate were also determined by ICP-MS. Leachate blanks were prepared by passing 100 mL of
deionised water through W41 filters that had been deployed in the aerosol sampler for 1 h. For example,
leachate blanks for Fe represented an average of $1.6 \pm 0.4$ % and $15.5 \pm 15.8$ % of the Fe sample
concentrations for GA03 and GA01, respectively). A subset of samples (GA03-2011) were also leached
using the instantaneous leach with freshly collected, filtered (0.2 µm) seawater as the leach medium.
Leachate blanks were subtracted from all leachate sample concentrations, details of which can be found in
Table S1 in the Supplementary Material.

The fractional solubility was calculated using Eq. (1):
$$\frac{[element]_{leach}}{[element]_{total}} * 100 = Fractional\ Solubility \tag{1}$$

Following the instantaneous UHP water leach, the filter was transferred to a 15 mL centrifuge tube, and
the second leach was undertaken, using 5 mL of 25 % (4.4 M) ultrapure acetic acid, with 0.02 M
hydroxylamine hydrochloride as the reducing agent (Berger et al., 2008). After a 10 min heating step (90
°C), the leaches were left for 24 h, before being centrifuged for 5 min at maximum power (3400xg). The
leachate was then carefully decanted into acid-clean LDPE bottles. In order to rinse any residual acetic
acid from the filter, 5 mL of UHP water was pipetted into the centrifuge tubes, which were then
centrifuged for a further 5 min on maximum power. This supernatant was then added to the relevant
leachate in the LDPE sample bottles. In this study, all samples were leached first using the UHP water
instantaneous leach, followed by a sequential leach with 25 % acetic acid. The overall solubility in 25%
acetic acid is calculated as the sum of the UHP water and acetic acid leaches divided by the total
concentration.

**2.4. Major anions and aerosol acidity**
Before the UHP water leachate was acidified, a 10 mL aliquot was taken from each sample leach for the
determination of the soluble major anions, $Cl^-$, $NO_3^-$ and $SO_4^{2-}$, by ion chromatography using either a
Dionex 4500i (at FSU for GA03 samples) or a Metrohm, IC850 system (at Laboratoire Interunivesitaire
des Systèmes Atmosphériques, Paris for GA01 samples). The aliquot was immediately frozen for storage.
Non-sea salt sulfate (nss-$SO_4^{2-}$) was calculated using the concentration of soluble $Cl^-$ as the reference
element to correct for $SO_4^{2-}$ from sea spray aerosols. In this study, aerosol acidity is estimated from the
concentration of $NO_3^-$ plus two times the concentration of nss-$SO_4^{2-}$ (Buck et al., 2010).






**2.5. Aerosol source characterisation**

Air mass back trajectory (AMBT) simulations were generated using the publicly-available NOAA Air

Resources Laboratory Hybrid Single-Particle Lagrangian Integrated Trajectory (HYSPLIT) model, using

the GDAS meteorology (Stein et al., 2015; Rolph, 2017). The 5-day AMBT simulations were used to

describe five regional categories, based on the predominant trajectories for the air masses. Simulations

and further details of these categories can be found in Wozniak et al., (2013; 2014) and Shelley et al.,

(2015; 2017). Briefly, for cruise GA03 air masses were characterised as European, North American,

North African, or Marine (no or minimal interaction with major continental land masses within the 5-day

simulation period). For cruise GA01, all the samples were classified as High Latitude dust (originating

north of 50°N; Bullard et al., 2016). The classifications are shown in Table S1.

240

**3. Results and Discussion**

**3.1. Total aerosol TEs**

Atmospheric inputs to the ocean are episodic, and exhibit a seasonality in the tropical and subtropical

North Atlantic that is largely driven by the migration of the intertropical convergence zone (Prospero et

al., 1981; Adams et al., 2012; Doherty et al., 2014). North African/Saharan mineral dust dominated

aerosol composition in the GA03 study region (Shelley et al., 2015; Conway and John, 2015; Conway et

al., submitted). In contrast, the GA01 transect was located north of the extent of the Saharan dust plume

(~ 25° N in summer, Ben-Ami et al., 2009), and was thus influenced by different, high latitude dust

sources (Prospero et al., 2012; Shelley et al., 2017), which also have a seasonal cycle. As a result, a large

dynamic range of aerosol loading was observed (Fe = 0.185–5650 ng m$^{-3}$; Al = 0.761- 7490 ng m$^{-3}$), with

the highest Fe and Al loadings associated with the North African samples (GA03), lower loadings with

the Marine samples (GA03), and the lowest loadings observed in the samples collected in the Labrador

Sea (GA01). Total aerosol TE data from the GA01 and GA03 cruises have been discussed in detail

elsewhere (Shelley et al. 2015; 2017), the data are only discussed here for comparison.

For all of the GA01 and GA03 samples, total Fe and Al were strongly correlated ($r^2$ = 0.999, Pearson's ρ

P < 0.01), demonstrating that the two metals have common lithogenic source(s) (Fig. 2). The correlation

between Fe and Al was largely driven by the heavily-loaded North African dust samples ($r^2$ = 0.997, P <

0.01). However, with the North African samples removed, only total Fe and Al from the GA01 High

Latitude dust ($r^2$ = 0.879, P < 0.01) and European ($r^2$ = 0.890, P < 0.05) samples were significantly

correlated, and there was no correlation between Fe and Al in the samples of  N. American ($r^2$ = 0.153, P





> 0.05) or Marine ($r^2$ = 0.0.16, P > 0.05) provenance (Fig. 2b) Looking more closely at the data, two of
the European samples (n = 4) have anomalously low Fe/Al ratios (Fig. 2; E3 = 0.48, E4 = 0.10), compared
to the other two samples of European origin (E1 = 0.95 and E2 = 0.78), and compared to the North
American samples, which are also thought to be strongly influenced by anthropogenic emissions (Fe/Al
1.1 ± 0.22, range 0.86-1.42).
Strong correlations for the combined data set including all samples were found between Ti/Al ($r^2$ = 0.999,
P < 0.01), Mn/Al ($r^2$ = 0.994, P < 0.01) and Co/Al ($r^2$ = 0.996, P < 0.01), in accord with previous
observations in this region owing to the primarily lithogenic source of these elements (e.g. Jickells et al.,
2016). The correlations between Al and the primarily anthropogenic TEs, Ni, Cu, Zn, Cd, and Pb, were
also significant at the 99% confidence level (Pearson's ρ): Ni/Al ($r^2$ = 0.884), Cu/Al ($r^2$ = 0.652), Pb/Al
($r^2$ = 0.478), Zn/Al ($r^2$ = 0.321), Cd/Al ($r^2$ = 0.303) and the fraction of the statistical variance accounted
for by the heavily-loaded North African samples ranges from 88% for Ni to 30% for Cd. Sources other
than mineral dust (e.g. metal smelting emissions, fly ash, vehicle emissions, volcanic ash, proglacial till)
are presumably responsible for the residual variance.
We have previously reported a mass ratio of 0.76 for Fe/Al for the North African end-member aerosols
(Shelley et al., 2015; Fig. 2a), which is significantly higher than the mean upper continental crustal (UCC)
ratio of 0.47 (Rudnick and Gao, 2003), suggesting that the North African aerosols were relatively depleted
in Al compared to Fe and other elements. Elemental mass ratios greater than the UCC ratio have
previously been observed for Saharan soils and dust (e.g. Guieu et al., 2002; Baker et al., 2013).While
there is evidence for anthropogenic source(s) of aerosol Fe to the North Atlantic (Fig. 2b), which has an
impact on aerosol Fe solubility (Sedwick et al., 2007; Sholkovitz et al., 2009; 2012), North African
mineral dust dominates the supply of Fe to much of the study region (Baker et al., 2013; Shelley et al.,
2015; 2017; Conway et al., submitted). In addition to the samples of European and North American
provenance, the Marine samples also show elevated Fe/Al ratios (Fig. 2b). The aerosols we classify as
"Marine" contain Fe and Al which presumably originated from continental source regions, but they may
also contain anthropogenic aerosols that have higher Fe/Al ratios. In addition, sea spray aerosols could
make a relatively higher contribution to the bulk aerosol in remote oceanic locations (de Leeuw et al.,
2014). However, this would have the opposite effect as the ratio of Fe/Al in surface seawater is two orders
of magnitude lower than the crustal ratio (0.017 − 0.024 in the North Atlantic gyre, 0.019 European
continental shelf, and 0.030 − 0.031 in the Mauritanian upwelling zone; Hatta et al., 2015). Hence the
contribution of sea spray aerosols appears to have a negligible impact on the Fe/Al ratios in the bulk
Marine aerosols.






### 3.2. Elemental mass ratios and aerosol provenance

Aluminium was used to normalise the aerosol loading data (Fig. 3). It was chosen instead of Ti, another proxy for mineral dust, due to the presence of some anomalously high Ti/Al ratios in some of the Marine samples during GA03 (Fig. 3a; Shelley et al., 2015). Due to the relative depletion of Al, relative to other TEs, in the North African aerosol samples collected during GA03, the elemental ratios reported here are higher than the UCC elemental ratios (Rudnick and Gao, 2003). Elemental mass ratios from the ten most heavily loaded GA03 North African aerosols were averaged to derive a value for the 'North African' ratio depicted by the dashed horizontal line in Figures 3(a-i). The similarity to the GA03 North African aerosols, which have an Fe/Al ratio of $0.78 \pm 0.03$ (Fig. 3c), and the relatively small range in the ratio of total Fe/Al for this same section ($0.85 \pm 0.20$), and evidence from the stable Fe ($\delta^{56}$Fe) fractionation in seawater and the aerosol UHP water leaches (Conway and John, 2014; Conway et al., submitted) provides evidence that North African mineral dust consistently dominates the supply of aerosol Fe to the tropical and sub-tropical North Atlantic.

In contrast, aerosols from the more northerly section, GA01, were largely outside the influence of the Saharan dust plume (Shelley et al., 2017), and are all classified as High Latitude (Fig. 3). During the first half of the cruise (samples G1-8, Fig. 1), the Fe/Al ratios were intermediate between the UCC ratio ($0.48 \pm 0.07$) and the North African mineral dust ratio ($0.78 \pm 0.03$) for both the total and soluble fractions (medians of 0.56 and 0.63, respectively). As the wind direction was predominantly from the north (Shelley et al., 2017), it is unlikely that the observed ratios reflect a mixture of North African mineral dust and European aerosols. Rather, it is more likely from a high latitude source, as dust supplied by proglacial till from Iceland and Greenland peaks in spring/early summer, and can be deposited over the Atlantic Ocean (Prospero et al., 2012; Bullard et al., 2016), although the extensive cloud cover experienced during the GA01 cruise (May/June 2014) prevented the use of satellite observations (e.g. http://worldview.earthdata.nasa.gov) which would have confirmed the presence of dust from this source. However, the TE concentrations reported by Achterberg et al. (2013) from volcanic ash sampled during the eruption of the Eyjafjallajökull volcano in 2011 offers some support for this argument, as our range of elemental ratios encompasses this end-member (Icelandic soils are almost exclusively volcanic in origin; Arnalds 2004), for most of the TEs, but also supports mixing with pollution-derived aerosols.

The most heterogeneous group of data were the samples from the most remote locations (Marine and High Latitude). This was also the group with the greatest difference in the Fe/Al ratio between the total and soluble fractions, and with the lowest ratios of Fe/Al in the soluble fraction (minimum Fe/Al = 0.15,



samples G9-GA01 and M3-GA03; Fig. 3c), suggesting that even though aerosol Fe is altered towards
more soluble forms during atmospheric transport (Longo et al., 2016), atmospheric processing renders Al
even more soluble relative to Fe.  However, there is some contradiction between the information from the
elemental ratios and the fractional solubility of Fe. Still using the examples of samples G9-GA01 and M3-
GA03 (low Fe/Al), both samples had European air mass back trajectories (Shelley et al., 2015; 2017).
However, the fractional solubility for Fe differed from 20 % for G9-GA01 to 0.8 % for M3-GA03,
suggesting that the reason for this disparity is that North African mineral dust was contributing to the
composition of the bulk aerosol during GA03. Recent advances in the determination of stable isotopes
present a powerful tool that can now be used to test such hypotheses, and have confirmed this was the
case on GA03, where there was a distinct isotopic signature associated with aerosols from Europe, and
North America, that differed from North African mineral dust (Conway et al., submitted).
For the anthropogenically enriched TEs, Ni, Cu, Zn, Cd and Pb (Figs. 3e-i) and for at least some of
samples of the mixed-source TEs, Mn and Co (Figs. 3b and d), there is some degree of source-dependence
in the elemental ratios, with some significant increases from the UCC mass ratios in the total (Shelley et
al., 2015) and UHP water soluble fractions (Fig. 3). The higher ratios of the UHP water soluble fraction
compared to the total indicates that these TEs are more labile than Al, which is related to where each TE
occurs on the particle (surface coatings versus matrix-bound). In addition, studies that have investigated
the size distribution of aerosols have found that pollution-derived TEs tend to be associated with fine
mode aerosols (< 1 µm diameter), which are more soluble than coarse mode aerosols due to the larger
surface area to volume ratio (Duce et al., 1991; Baker and Jickells 2006), some fraction of which will pass
through the 0.2 µm filter.  Size fractionated samples were collected during the GA03 cruise, and the
smaller size fractions were indeed more soluble than the larger (Landing and Shelley, 2013). Enrichment
of TEs with predominantly anthropogenic sources accords with other studies in the North Atlantic, and is
most striking for aerosols that did not originate from the sparsely-populated arid regions of North Africa
(e.g. Buck et al., 2010; Gelado-Cabellero et al., 2012; Patey et al., 2015; Shelley et al., 2015).
In addition, positive matrix factorisation analysis suggests that aerosols from this study were dominated
by two factors, a mineral dust factor and a pollution factor (Fig. S2a, Supplementary Material).
Unsurprisingly, the mineral dust factor dominated where North African aerosols were sampled, and the
pollution factor dominated closer to the European and North American continents (Fig. S1b). This is in
accord with the samples from North Africa having elemental mass ratios that are consistently the closest
to the UCC elemental ratios compared to aerosols from the other source regions (Fig. 3).  In the High
Latitude samples, the pollution factor and the mineral dust factor were of approximately equal dominance.



Interestingly, the North African aerosols also contained a relatively strong pollution component,
consistent with a northeast flow into North Africa from Europe, followed by entrainment of mineral dust
during passage over the Sahara. Given that the PMF indicates that 100 % of the variability in the Cd
concentrations was explained by the pollution factor, this suggests that Cd in North African aerosols is not
sourced from mineral dust, which would explain why no fractionation was observed in Cd isotopes from
North African and European aerosols (Bridgestock et al., 2017). Further, it also suggests that even the
relatively homogeneous aerosols of North African provenance do not represent a 'pure' end-member.

**3.3. Aerosol solubility**
**3.3.1. Solubility of aerosol Fe and Al: UHP water (instantaneous) compared to 25 % acetic acid**
**leaches**
The UHP water soluble fraction of aerosol Fe and Al determined for all the North Atlantic GA01 and
GA03 samples varied by two orders of magnitude (Fig. 4a: Fe = 0.14 - 21 %, median 2.2 %; Fig. 4c: Al =
0.34 - 28 %, median 2.7%). Although a broader range of Fe and Al solubility was observed in this study,
both these results and those reported by Buck et al. (2010) using the same approach (Fe = 2.9 - 47%,
median = 14%, and Al = 3.7 - 50%, median = 9.5%) broadly agree that the median UHP water soluble
fractions of Fe compared to Al in the North Atlantic are similar. While there was considerable overlap in
the ranges of fraction solubility of TEs in aerosols of different provenance (e.g. Fe: European 1.9 – 21 %;
N. American 0.84 – 8.8 %; Marine 1.7 –18 %; High Latitude dust 1.9 – 20 %), the North African samples,
identified by their orange colour, high Fe and Al loadings, and definitive air mass back trajectories)
formed a distinct cluster of very poorly soluble Fe, or Al (< 1%; Fig. 4a and c). However, the solubility of
the North African ('Saharan') aerosol Fe was 1 – 2 orders of magnitude lower in this study (0.14 – 0.57
%) than during the Buck et al. (2010) study (2.9 – 19 %). This supports the hypothesis that TEs from
North African aerosols sampled with a higher frequency closer to source (as in this study) are less soluble
as a result of a lesser degree of atmospheric processing and/or larger particle sizes (Baker and Jickells,
2006; Longo et al., 2016).  Furthermore, given that the Sahara Desert is the largest source of mineral dust
to the atmosphere globally (the North Atlantic Ocean receives ~ 40 % of the mineral dust inputs to the
global ocean, Jickells et al., 2005), the effects of increasing industrialisation/urbanisation of African
countries, coupled with large unknowns in the magnitude of future mineral dust supply, and biomass
burning, the ability of models to replicate subtleties in aerosol TE solubility may prove critical in
forecasting ecosystem impacts and responses. In other words, it is important to accurately constrain
aerosol trace element solubility with high quality data in order to improve the predictive capacity of
models.



The relationship between total aerosol Fe and the soluble fraction can be described by a hyperbolic
function (Fig. 4a and b), in accord with Sholkovitz et al. (2009; 2012). The same relationship was
observed for aerosol Al (Fig. 4c and d). The insets in Figure 4 plot the data on a log-log scale, and
illustrate the inverse relationship between Fe or Al solubility and atmospheric loading, and demonstrates
that while the absolute values for solubility are dependent on the leach media used, the general trend is
maintained. Jickells et al. (2016) compiled solubility data from the North Atlantic and also found that the
inverse relationship between Fe and Al solubility and atmospheric loading was robust over the range of
atmospheric loadings found in the North Atlantic, regardless of the leach protocol employed.
In this study, both the UHP soluble, and 25 % acetic acid soluble fractions of Fe and Al (Figs 4a and d)
were related to atmospheric loading, i.e. the highest loaded North African samples had the lowest
solubility.  This trend is consistent with the observations of Jickells et al. (2016). The possible exception
to this trend is the fraction of Al that dissolved from North African aerosols following the 25 % acetic
acid leach (Fig. 4d). However, it could simply be that we are observing scatter in our data, which is
smoothed out in the larger dataset (n > 2000) examined by Jickells et al. (2016). Furthermore, the
solubility range estimated for Al (0.3 – 28 % and 4.1 – 100 %  solubility in UHP water and 25 % acetic
acid, respectively) far exceeds the relatively narrow range used in the MADCOW model (1.5 – 5 %),
which has been used to estimate atmospheric inputs based on dissolved Al concentrations in the mixed
layer (Measures and Brown, 1996), even taking the more conservative UHP water-derived values (0.3 –
28 %), which has implications for the estimation of atmospheric deposition fluxes. It is noted, however,
that the median values from this study fall within the range used by the MADCOW model (2.7 % and 3.3
% for UHP water and 25 % acetic acid, respectively).
Furthermore, the fractional solubility of bioactive TEs from aerosols is taken as a rough approximation of
bioavailability. Yet, clearly the choice of leach media impacts the estimated value. For elements with
generally low solubility, such as Fe, the difference between 1 % and 2 % solubility is an increase of 100
%, meaning that only half the amount of dust is needed to yield the same amount of dissolved Fe, the
most-readily bioavailable form of Fe (Shaked and Lis, 2009). To complicate matters further, recent
research has demonstrated that some diazotrophs are able to directly access particulate Fe (Rubin et al.,
2011). Given that the different leaching approaches access different fractions of TEs (loosely bound to
surfaces compared with associated with less reactive phases), that dissolve from aerosols at different rates
(e.g. Kocak et al., 2007; Mackey et al., 2015), we need to conduct experiments that elucidate the
relationship between the soluble and bioavailable fractions.

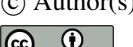



Furthermore, in productive regions where Fe (and perhaps other TEs) associated with lithogenic particles
is directly available to micro-organisms (Rubin et al., 2011) or in regions where particulate Fe is
processed by zooplankton (Schmidt et al., 2016), we are likely underestimating bioavailable Fe using the
instantaneous approach (Buck et al., 2006). Therefore, the different leaching approaches can be used to
probe specific questions related to the response of the microbial community to changes in aerosol type
and/or supply by providing a "window of solubility". This is an important consideration as the true
solubility of aerosol TEs in the upper ocean cannot be directly measured.
**3.3.2. Solubility of TEs: UHP water (instantaneous) compared to 25 % acetic acid leaches**
All ten TEs from the five different provenances were less soluble in UHP water than 25 % acetic acid
(Fig. 5). This is not a surprising finding given the lower pH of acetic acid compared with UHP water, that
acetate is a bidentate ligand, and the longer contact time of the aerosols with the leach solution in the 25
% acetic acid leach procedure. In addition, there is some degree of source-dependent variability in the
relative proportions of each TE that is released by the two leaches. In general, as with the leaches with
UHP water, the North African aerosols were distinctly less soluble than aerosols from the other source
regions (Fig. 5). Figure 6 highlights the distinction between the lithogenic elements, Al, Fe and Ti
(universally low solubility in UHP water, mostly < 20 %, and extremely low solubility of North African
aerosols, < 1 %) and the pollution-dominated elements, Ni, Cu, Zn, Cd and Pb (solubility up to 100 %).
Manganese (Mn) and Co have both lithogenic and anthropogenic sources (mixed-source), and have
intermediate solubilities. Like all the TEs reported here, Mn solubility in UHP water was significantly less
($p < 0.01$, two-tailed, homoscedastic t-test) in North African aerosols (median solubility = 19 %) than in
the non-North African samples (median = 38%), which seems to contrast somewhat with the findings of
Baker et al. (2006) and Jickells et al. (2016). However, in common with these earlier studies (Baker et al.,
2006; Jickells et al., 2016), there was no significant source-dependent difference in Mn solubility in 25 %
acetic acid (non-North African samples: $49 \pm 15\%$, North African samples: $49 \pm 6.4\%$). The differences
source dependence of Mn solubility in these aerosols between the two leach types is just one example of
how challenging it is to model Mn bioavailability in the North Atlantic. Nevertheless, with the exception
of soluble Mn from the acetic acid leach, this TE solubility data supports the general assertion that aerosol
TE solubility varies as a function of provenance and/or atmospheric loading (e.g. Baker and Jickells,
2006; Sedwick et al., 2007; Sholkovitz et al., 2009; 2012; Aguilar-Islas et al., 2010; Fishwick et al., 2014;
Jickells et al., 2016).
**3.3.3. Soluble TEs: UHP water compared to seawater instantaneous leaches**
Seawater leaches were conducted on a subset of samples (GA03-2011), to investigate the suitability of
seawater as the leach medium in the instantaneous leach. During this study, Fe solubility in seawater was



lower than in UHP water (Fig. 6c). This phenomenon has previously been observed in atmospheric
aerosols from the North Atlantic Ocean (Buck et al., 2010).  For Fe, only a few samples of North
American and Marine provenance conformed to the relationship described by the equation proposed by
Buck et al. (2010), with most of our data plotting above the regression line of the Buck et al. (2010) study
(Fig. 6c), indicating that our data was relatively more soluble in UHP water compared to seawater than in
this earlier study. One possibility is that the higher aerosol Fe loadings we observed during GA03-2011
(this study, maximum = 5650 ng Fe m$^{-3}$), compared to the A16N-2003 transect (Buck et al. 2010;
maximum =1330 ng Fe m$^{-3}$), resulted in a particle concentration effect (Baker and Jickells, 2006),
whereby the relationship between aerosol Fe loading and fractional solubility breaks down because  dust
on the filter can be a source of soluble Fe but can also scavenge dissolved Fe from the sea water leach
solution as it passes through the filter.  An alternative explanation for the difference in Fe solubility is that
the organic composition of the seawater used as the leach mediums differed between the two studies,
given that the link between Fe solubility in seawater and Fe-binding ligand availability is well established
(e.g. Rue and Bruland, 1995; Gledhill and Buck, 2012).
Mn is the only TE that has a slope close to unity (0.98; Fig. 6b), suggesting that solubility estimates were
not impacted by the choice of leach medium used. This is consistent with other studies that have found
that Mn solubility is less sensitive to the choice of leach media, or to aerosol provenance than other TEs
(Baker et al., 2006; Jickells et al., 2016). Due to the large variability in the data, there was no significant
difference between Mn solubility in UHP water or seawater (32 ± 13 % and 24 ± 17 %, respectively; Fig.
S1, and Tables S2 and S3, Supplementary Material). Other TEs that also resulted in similar solubility
estimates were Al, Cu, Zn and Cd (Figs 6a, f, g and h). However, the data also had high variance,
particularly Al, so caution is urged in interpreting this as a 1:1 relationship. Indeed, an ANOVA indicated
that the means of the UHP water and seawater leaches were equal for each element at the 95 %
confidence level. When the data is considered by aerosol source, however, there were some source
dependent differences between the two leaches for Al (North American, Marine and North African), Fe
(North African), Co (Marine), Zn (North African), and Cd (North African). The aerosol populations with
significant differences between the means are indicated in the brackets.
For Co and Pb (Figs 6d and h), most of the data falls below the 1:1 line, indicating that they were
generally more soluble in seawater than UHP water. In contrast, the opposite trend was observed for Fe
and Ni (Figs 6c and e), pointing to differences in the availability of metal binding ligands in the seawater
used. A challenge of using seawater as the leach medium is that it is difficult to control for natural
variability in its organic composition. Consequently, it is not possible to determine conclusively why



contrasting trends in the fractional solubility of TEs were observed. For this reason, we advocate for the
use of UHP water as a common leach medium to facilitate comparisons of solubility resulting from
differences in aerosol composition. An additional benefit is the ease of analysis of UHP water compared
to seawater.  That is not to say that seawater should not be used, but rather that it is difficult to draw direct
comparisons between datasets due to potential differences in the composition of the seawater used, which
could affect the fractional solubility.

### 3.4. Aerosol acidity

Numerous laboratory studies have demonstrated a link between atmospheric acid species and Fe solubility
(e.g. Spokes and Jickells, 1995; Desboeufs et al., 1999; Meskhidze et al., 2003), and in field studies some
degree of correlation between nss-$SO_4^{2-}$ and soluble Fe has been observed for samples collected over the
Pacific (e.g. Hand et al, 2004; Buck et al., 2013) and Atlantic Oceans (Johansen et al., 2000). However, in
other studies in the Atlantic Ocean (Baker et al., 2006; Buck et al., 2010) no relationship was observed.
Similarly, we observed no correlation between the soluble acid species, $NO_3^-$ and nss-$SO_4^{2-}$, and the
percentage of UHP water soluble Fe ($r^2$ = 0.056 and 0.005, respectively). There was also no significant
correlation between the percentage of UHP water soluble Fe and aerosol acidity ($[NO_3^-] + 2*[nss-SO_4^{2-}]$)
for the GA03 samples ($r^2$ = 0.08; Fig. 7a), which further suggests the dominance of North African mineral
aerosol, which does not have a large $NO_3^-$ or nss-$SO_4^{2-}$ component (Baker et al., 2006). As acid species
and Fe predominantly reside in different size fractions of aerosols (Schulz et al., 1998; Raes et al., 2000),
an inverse relationship or no relationship could be a result of a low degree of internal mixing in the
aerosol samples, as opposed to aerosol acidity not exerting any control on Fe solubility (Baker et al.,
2006), whereas a positive correlation suggests that aerosol acidity is exerting a control on Fe solubility.
Further investigation of the GA03 aerosols, split into their provenance categories, suggests that there was
little effect of aerosol acidity on Fe solubility for the North American, Marine or North African samples.
The European samples also showed no clear trends between the four, uncorrelated data points. In contrast,
the weak positive trend in the GA01 samples (High Latitude dust; $r^2$ = 0.52; Fig. 7b) could be a kinetic
effect resulting from aerosol processing at high altitudes.
During their investigations of the GA03 aerosols, Wozniak et al., (2013) proposed a role for water soluble
organic carbon (WSOC) in controlling the solubility of Fe. Desboeufs et al. (2005) also found evidence
for a link between total carbon and TE solubility in regions impacted by anthropogenic activity. Thus,
both aerosol acidity and organic carbon content are implicated as controls on aerosol Fe solubility, but the
relationship is frequently not linear. One explanation for this lack of linearity was proposed recently by
Hennigan et al. (2015). They concluded that molar (or mass) ratio techniques are not suitable for





predicting aerosol pH (or acidity), and cautioned against drawing conclusions based on proxy methods
(e.g. nss-$SO_4^{2-}$ or $NO_3^-$). Instead, they recommend that either a thermodynamic modelling approach
(constrained by gas and aerosol measurements), or the phase partitioning of $NH_3$, should be used for
predicting aerosol pH. Weber et al. (2016) take this further and argue that the best approach for predicting
aerosol pH is the phase partitioning of $NH_3$. These approaches are beyond the scope of the present study,
but should be a consideration for future studies due to the pH-dependency of aerosol TE dissolution,
especially given the neutralising influence of $NH_3$, carbonate mineral phases, and sea salt.

### 4. Conclusions

In this study, five potential aerosol sources were identified based on air mass back trajectory simulations;
i) North African, ii) European, iii) North American, iv) High Latitude, and v) Marine. Of these five
categories, the North African aerosols were the most homogeneous in terms of their fractional solubility
and elemental mass ratios. In contrast, samples from the most remote locations, the Marine and High
Latitude aerosols, were the most heterogeneous. Elemental ratios were presented rather than enrichment
factors, as earlier work highlighted that the UCC ratios are not representative of the North African mineral
dust end-member.
As TE solubility cannot be directly measured, biogeochemical models require a robust relationship
between two or more parameters that can be used to predict TE solubility in order to constrain the
bioavailable fraction of TEs. However, in regions of high mineral dust deposition and/or productivity
fractional solubility (bioavailability by proxy) we are likely to be underestimate solubility using the
instantaneous leach approach. As previously reported, we observed an inverse relationship between TE
fractional solubility and aerosol provenance/loading for all leach media (UHP water, filtered seawater,
and 25 % acetic acid) investigated, with the exception of Mn. However, the large degree of variability in
the data meant that few of these differences were statistically significant. There were also differences in
the solubility estimates calculated from the different leaches, with values derived from the 25 % acetic
acid leach always highest, by approximately an order of magnitude. Leaches conducted using filtered
seawater resulted in the lowest values for TE solubility, except for Pb, which was more soluble in
seawater than UHP water. Such differences, serve as a reminder that some degree of standardisation is
required for aerosol leach protocols, to facilitate comparisons between different studies. This will be key
moving forward if TE solubility is to be accurately parametrised in biogeochemical models. Further work
is also required to assess which fraction is accessed by the various leach protocols in order to elucidate
links between the soluble and bioavailable fractions.




**Data availability**

Data is available at BCO-DMO (GA03; www.bco-dmo.org) and LEFE-CYBER (GA01;

(www.obsvlfr.fr/proof/php/GEOVIDE/GEOVIDE.php), and on request from the lead author.

**Acknowledgements**

Many thanks to the captains and crews of the RV Knorr (GA03-2010 and 2011) and NO Pourquoi Pas?

(GA01), the chief scientists (GA03 = Bob Anderson, Ed Boyle, Greg Cutter; GA01 = Geraldine Sarthou

and Pascale Lherminier), Alex Baker for the loan of the aerosol sampler used on GA01, and Alina Ebling

Petroc Shelley, Alex Landing and Sarah Huff for their help running samples. This work was supported by

grants to WML (NSF-OCE 0752832, 0929919 and 1132766), and GS (ANR-13-B506-0014 and ANR-12-

PDOC-0025-01). RUS was supported by a LabexMER International Postdoctoral Fellowship and CG29

Postdoctoral Fellowship. A portion of this work was performed at the National High Magnetic Field

Laboratory, which is supported by National Science Foundation Cooperative Agreement No. DMR-

1157490 and the State of Florida. The aerosol digestions for GA01 were undertaken in the geochemistry

clean room at Ifremer (Centre de Bretagne). Trace element determination for GA01 was conducted at the

Pôle de Spectrométrie Océan at the Institut Universitaire Européen de la Mer with the support and

guidance of Claire Bollinger and Marie-Laure Rouget.

567

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





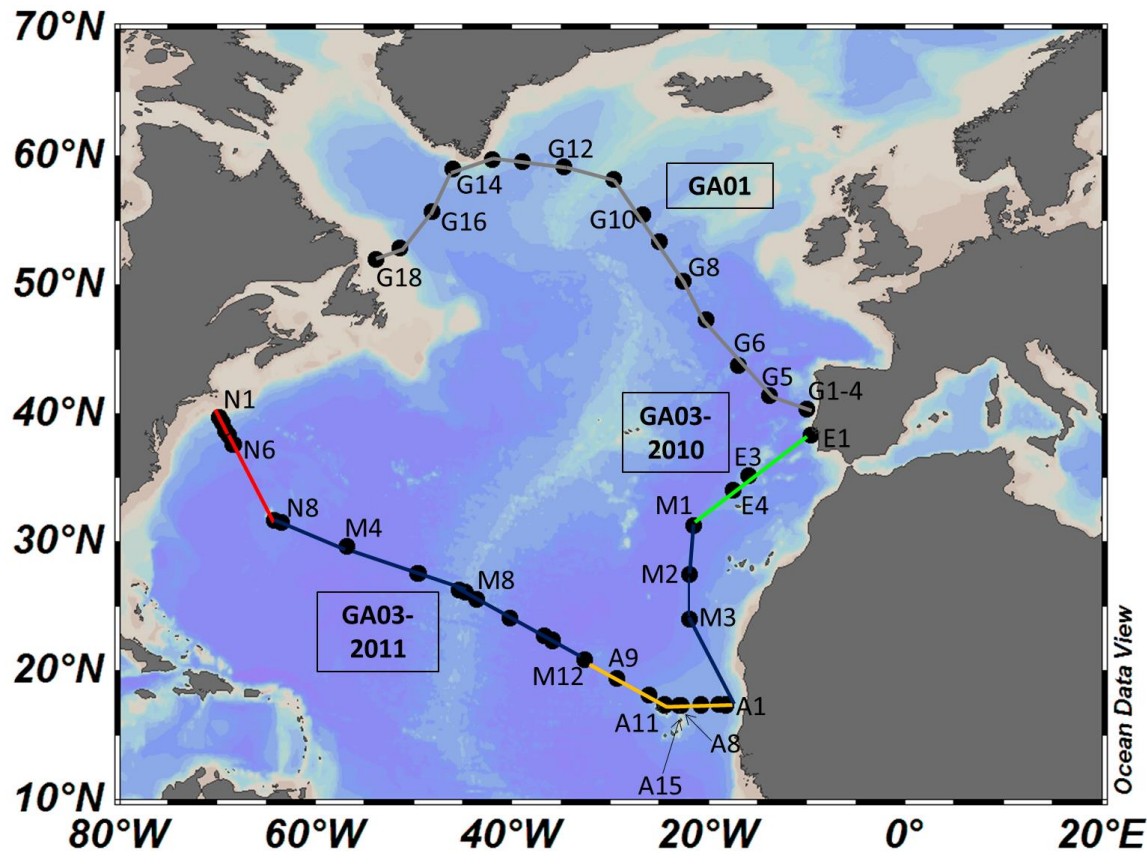


**Figure 1**


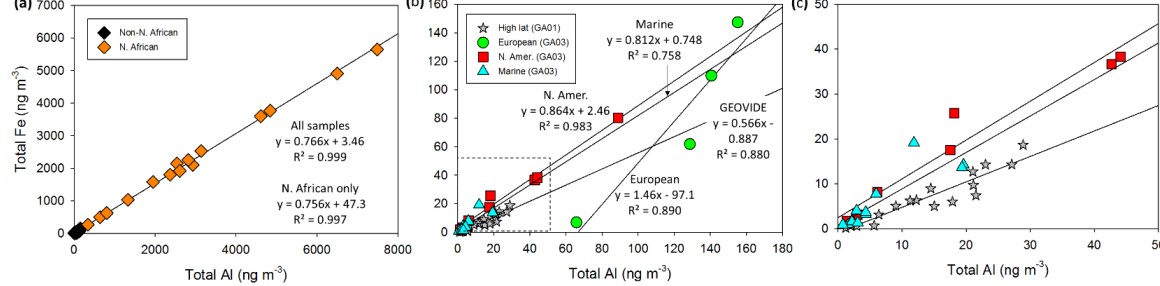


**Figure 2**




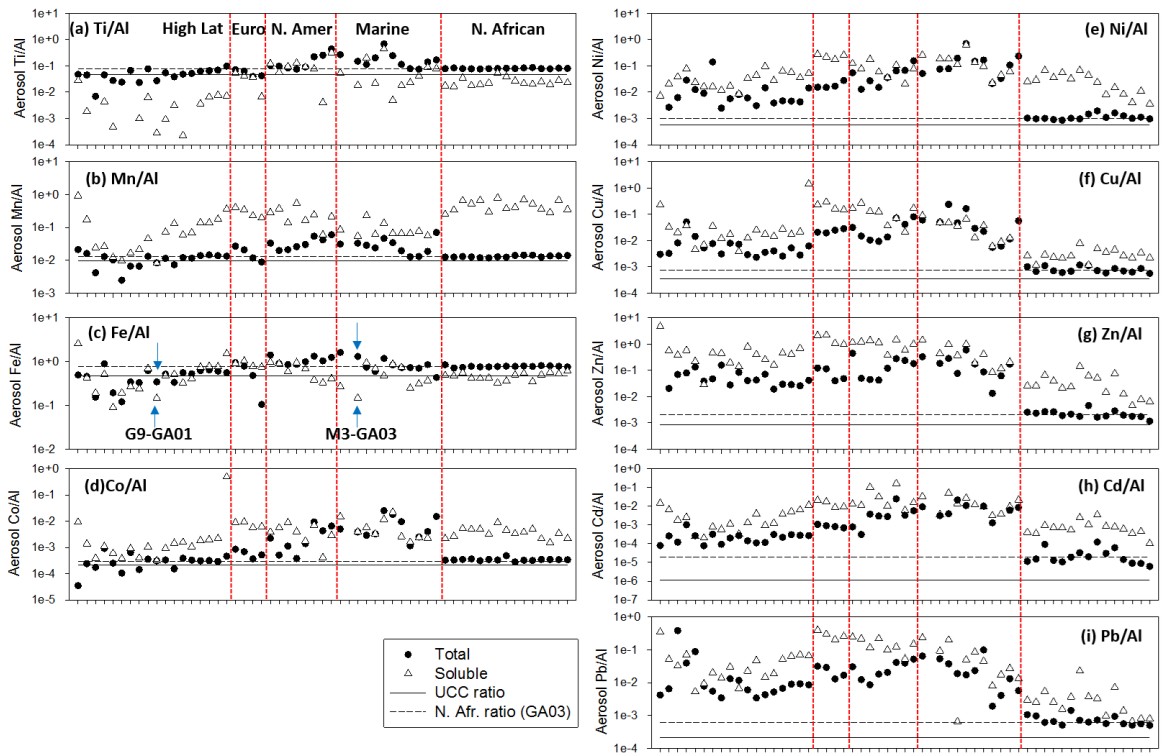

**Figure 3**







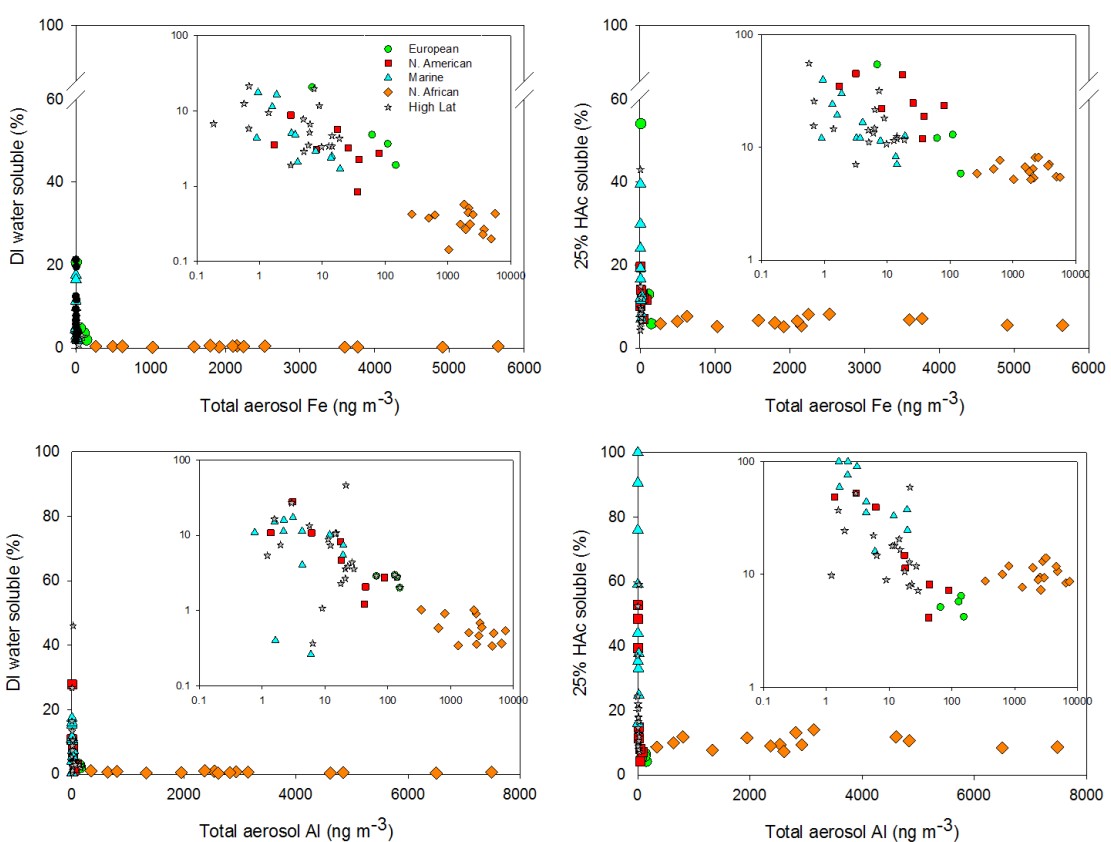


**Figure 4**







**Figure 5**





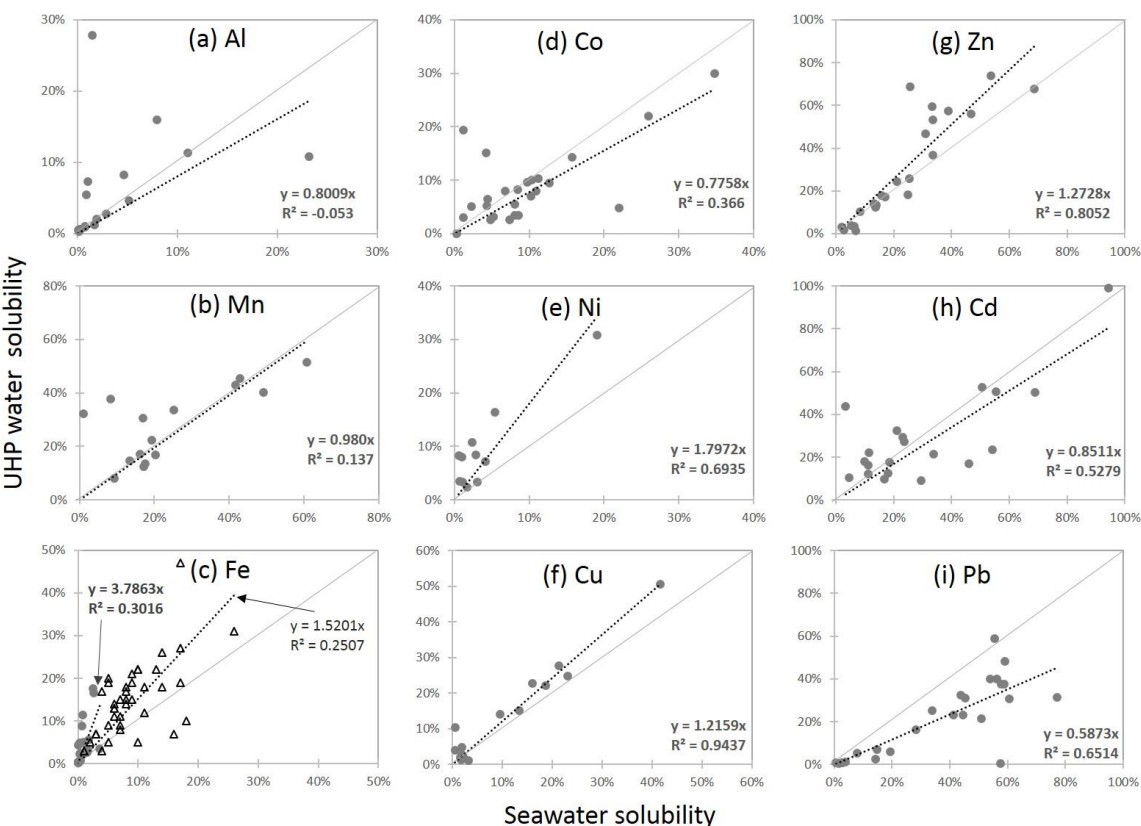


**Figure 6**







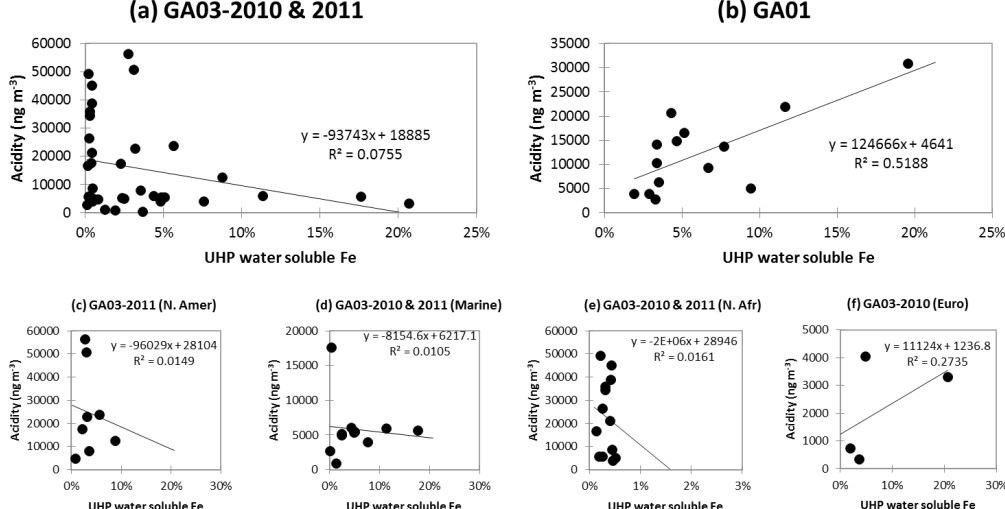

**Figure 7**

**Captions**

**Figure 1. The GEOTRACES GA01 and GA03 cruise tracks (GA01, GA03-2010 and GA03-2011). In total, 57**
**aerosol samples (GA01 n = 18, GA03 n = 39; black dots) were collected. The samples are grouped by aerosol**
**provenance (green = European (E1-4), blue = Marine (M1-12), yellow = North African (A1-15), red = North**
**American (N1-8), and grey = High Latitude (G1-18)), identified from air mass back trajectory simulations**
**using the NOAA ARL model, HYSPLIT (Stein et al., 2015; Rolph, 2017).**

**Figure 2. Total aerosol Fe and Al (ng m$^{-3}$) for: (a) all aerosol samples from cruises GA01 and GA03, (b)**
**samples from sources other than North Africa (i.e. the black diamonds in Fig. 2a), and (c) the samples inside**
**the dashed box in Fig. 2b. For High Latitude dust n = 18, European samples n = 4, North American samples**
**n = 8, Marine samples n = 12, and Saharan samples n = 15.**

**Figure 3. Elemental mass ratios (normalised to Al) of total (black circles) and UHP water soluble (white**
**triangles) TEs. The UCC elemental ratio (Rudnick and Gao, 2003) is indicated by the solid horizontal line,**
**and the elemental ratio in North African sourced aerosols (Shelley et al., 2015) is indicated by the dashed**
**horizontal line on each plot. The red vertical lines separate the aerosol source regions, which are labelled in**
**panel (a). Samples G9-GA01 and M3-GA03 are indicated by blue arrows in panel c (see text for details).**

**Figure 4. (a) Percentage of UHP water soluble Fe versus total aerosol Fe (ng m$^{-3}$), (b) percentage of 25 %**
**acetic acid soluble Fe versus total aerosol Fe (ng m$^{-3}$), (c) percentage of UHP water soluble Al versus total**
**aerosol Al (ng m$^{-3}$), and (d) percentage of 25 % acetic acid soluble Al versus total aerosol Al (ng m$^{-3}$).  the**
**percentage of soluble Fe, or Al, versus total Fe, or Al, is described by a hyperbolic function (Sholkovitz et al.,**
**2009; 2012). The insets in each panel plot the same data on log-log scales to demonstrate the inverse**
**relationship between the two parameters.**

**Figure 5. Solubility of Al, Ti, Mn, Fe, Co, Ni, Cu, Zn, Cd, Pb following a UHP water leach (UHP water, black**
**circles), and a sequential leach of 25 % acetic acid (HAc, grey squares). The red vertical dashed lines**
**represent the different aerosol source categories, as labelled in panel (b). Note that Ti (panel b) is highly**
**insoluble and the y axis has a maximum value of 20%.**



**Figure 6. Comparison of TE solubility following instantaneous leaches using UHP water or locally-collected,**
**filtered seawater. The solid line is the 1:1 line. Where fewer data are observed, concentrations were below**
**detection for one or both of the two leaches. The data from Buck et al. (2010) is plotted as open triangles in**
**panel (c).**
**Figure 7. Aerosol acidity versus UHP water soluble Fe (%) for (a) GA03-2010 and 2011, (b) GA01 – High**
**Latitude dust, (c) GA03 – N. American, (d) GA03 – Marine, (e) GA03 – North African, and (f) GA03 –**
**European samples.**