# Peer review of "Regional trends in the fractional solubility of Fe and other metals North"

_Biogeosciences, 2017_

## Referee Comment (RC1) · Anonymous Referee #1 · 12 Dec 2017

This manuscript describes the results obtained from different leaching methods used to liberate soluble trace elements from aerosol samples collected over the North Atlantic Ocean. The majority of the work focuses on leaches with ultra-high purity water (UHP) and an acetic acid solution, although some results obtained from leaching with seawater are also reported. The dataset presented is of high quality and it has been subjected to a very thorough analysis. The manuscript is highly suitable for publication in Biogeosciences, although I feel that some clarifications of relatively minor points are

necessary.

Although the manuscript focuses on the UHP and acetic acid leach results, Section 3.3.3 presents results obtained by leaching a subset of the aerosol samples using seawater. Currently the manuscript contains no information about how these experiments were performed. Please add this.

Non-seasalt (nss) sulfate concentrations in the aerosol samples were estimated using aerosol chloride concentrations as an indicator of seaspray content. I understand that this was the only indicator available, since major cations (e.g. sodium or magnesium ion concentrations) were not measured. However, chloride is not an inert species in marine aerosol, since it can be converted to hydrogen chloride and lost to the gas phase (Andreae and Crutzen, 1997). The extent of chloride loss is likely to be greater in polluted air types, such as those originating over Europe and North America, but all of the samples will be affected to some extent. Thus the nss concentrations presented here will be over-estimated and it would be helpful to note this in the text.

I found it difficult to compare the results presented in the main body of the manuscript with those in the Supplementary Information (and in the two earlier manuscripts in which previous work on these samples was presented) because different labels have been used for the samples in different places. If it is not possible to only use a single set of labels, please could all the different labels be added to the tables in the supplement?

There is also a specific issue with the naming convention for sample M3-GA03. The "M" here (and the map in Shelley et al., 2015) imply that this was a Marine sample. The discussion on lines 328 -332 specifically state that it had a European air mass back trajectory. Please clarify this.

I assume that positive matrix factorisation analysis (lines 350 onward) was done using total trace element concentrations. Please could this (or the correct information) be specifically stated.

All of Figures 2 – 7 would be improved by the addition of error bars. This would greatly assist the reader in putting the relatively high variability in calculated parameters (TE ratios to Al, or percentage soluble fractions) for samples with low total concentrations into context with the low variability, high total concentration samples with North African origin.

Minor points:

Line 95: "seawater" spelling.

Lines 103-115. Much of this paragraph is repetition of material from previous paragraphs. It could easily be shortened.

Line 319: The eruption of Eyjafjallajökull took place in 2010, not 2011.

Line 353: I think Fig S2b should be referenced here, not S1b.

Line 369: The panels of Fig. 4 are not labelled on the figure.

Line 374: "the ranges of fraction solubility" - I think there is a misspelling here.

Lines 382-387: This is a very long and cumbersome sentence. Please consider splitting it.

Lines 444-445: "The differences source dependence of" Please correct.

Line 481: Panel h of Fig 6 shows data for cadmium. Lead data is on panel i.

Lines 509-511: This statement is very speculative. Please add further explanation or consider removing.

Line 534: I am a little confused by the opening statement of this sentence. This manuscript has been devoted to the direct measurement of TE solubility!

Lines 536-537: "in regions of high mineral dust deposition and/or productivity fractional solubility". An odd construction. What is productivity fractional solubility?

Lines 538-839: How is it possible to have an inverse relationship between TE fractional solubility and aerosol provenance?

References Andreae, M. O., and Crutzen, P. J.: Atmospheric aerosols: Biogeochemical sources and role in atmospheric chemistry, Science, 276, 1052-1058, 1997.

---

## Referee Comment (RC2) · K. Desboeufs (Referee) · 15 Dec 2017

General comments:

This paper reports Fe, Al and trace metals composition and solubility in aerosols particles collected in North Atlantic Ocean. Even if it exists a lot of data on the solubility of Fe and major trace metals in Atlantic area (for ex. Clivar, AMT.. campaigns), new data and in particular new perspective for using this data is always interesting. In this

idea, the title of this manuscript was promising but finally, it is rather disappointing after reading. Indeed, even if the title implies a discussion on the relation between solubility and aerosol origin, the paper is limited to a dataset of solubility values as a function of air mass back-trajectories. It's very frustrating because the material is present to make this study original and significant. Another purpose of paper is the investigation to different leaching protocol for solubility measurement standardization. This twofold topic (provenance and protocol) makes abstruse the main objective of the paper.

Although this experimental work has been carefully conducted and contains interesting results, it is short of new findings in this current state due to a lack of guidance in the discussion on the results and of comparison with the literature. Consequently, I strongly encourage the authors to work again on this manuscript because your data could bring appreciable and consistent results for the community.

The major highlights proposed in this paper are "trace elements from aerosols from 1) North Africa were always the least soluble, and the most homogeneous . . ., 2) aerosols from the most remote locations were generally the most soluble, but had the most spread in the values of fractional solubility and 3) primarily pollution-derived TEs (Ni, Cu, Zn, Cd and Pb) were significantly enriched above crustal values in aerosols, even in samples of North African origin."

The critical point on this paper is the signification of term "provenance". In the manuscript, the aerosol provenance is considered as the "back-trajectory". Firstly, the back-trajectories were made for an arrival height of 500m, whereas the maximum altitude of boundary layer is often between 200 and 600 m in North Atlantic (Petenko et al., 1996; Fuhlbrügge et al., 2013), and aerosol sampling was in the boundary layer, so it is not automatically consistent. Moreover, the "loading" in various aerosols is not only dependent on the trajectory but also on emissions along the transport of air masses. A same air masses can be a mix between various types of aerosols. In consequent, as solubility is dependent on kind of particles, it's obvious to observe a largest variability of solubility for aerosols from most remote locations in comparison to dust samples

(findings 1) and 2)). In one case, it's a same source and in the other case that includes various sources and hence kind of particles. The relevant conclusion of this work should precisely be that aerosol provenance is not sufficient to estimate composition and solubility of trace metals.

From your database and the previous conclusions of Shelley et al. (2015) on the identification of sources of TEs (not provenance) in aerosols, further investigations could enable to emphasize a relationship between TEs solubility and origin. In this purpose, the origin of aerosol is a huge question which demands a more extensive bibliography to provide convincing proof of metals sources. A part of this comparison with bibliography is provided in the session 3.1 and 3.2. Nonetheless, the structuration of the paper is to much confuse to extract the pertinent information (see specific comments). These sessions should only include a summary of appropriate results from Shelley et al. (2015) for the discussion on the link with solubility. Moreover, a relevant bibliography is often missing in the manuscript. Several field campaigns during the last decade (AMMA, DODO, DABEX or SAMUM) + specific works (e.g. Trapp et al., 2010) permitted to improve our knowledge about the African dust composition before and after transport. Even if this literature is focused on major metals as Al, Fe or Ti, this literature about Saharan dust characterisation should be used to validate/discuss your aerosol origin. Moreover, the category "high latitude dust" is very extended with Al/Fe ratio ranging from 0.1 to 1 (L322-334), suggesting a variability of aerosol sources in these samples (see specific comments). The conclusion of authors about the mixed volcanic and anthropogenic origins of GA01 samples is supported only by the back-trajectories (not shown) and a comparison with the TEs concentrations in volcanic ash from Achterberg et al. (2013), but no comparison on elemental ratios is provided. Is this variability is consistent with the typical composition of volcanic or cold environment dust? Again, several papers report Fe or metals content in high latitude volcanic regions producing dust as Iceland (e.g. Baratoux et al., 2011, Óladóttir et al., 2008 and 2011). It could be interesting to compare with these data for discriminating the origin of this high latitude dust.

The third conclusion of a paper is that polluted derived TEs were significantly enriched above crustal values in aerosol and notably in dust samples (A1-15), on the basis of comparison with UCC. This conclusion is already in Shelley et al. (2015) for GA03 cruises. The new information could be for GA01 for high latitude dust, but this work is not done. Furthermore, I'm not agree with your conclusion on the enrichment "even in samples of North African origin". As a matter of facts, Shelley et al. (2015) conclude that except Cd, the "pollution-derived" elements present EFs less than 10 with respect to Al in the African samples (See Shelley et al., 2015, figure 6). Moreover, your main argument is that these elements have some significant increase from the UCC mass ratio in your dust samples (see specific comments: P11,L336). However, it's known that the production of mineral dust is associated to a chemical fractioning due to the size partitioning between coarse rich-Si grains and the finest clay fraction during sand-blasting. It means an enrichment in Al, Fe, Ca.. in dust particles in comparison to soil (Lafon et al., 2006). Thus, the authors are surprised at Al/Fe ratio around 0.76 (P9, L275). Yet, this value is fully consistent with the common knowledge on African dust (see previous comment on bibliography + e.g. Formenti et al., 2011 or 2014 and ref therein or Lazaro et al., 2008). Due to the trace metals partitioning in soil size fraction, this fractioning and hence enrichment probably happens also for trace metals in mineral dust. Your data seem to confirm this trend, previously observed by Trapp et al. (2010) after long-range transport. it's bad that this point of view is not addressed here. . .

Finally, a part of the manuscript is focused on the standardization of methods to estimate TE solubility. The conclusion on this session is not convincing for me because the proposed "upper limit" is not supported by the "reality" of aerosol dissolution in atmospheric or sea waters. The 25% acetic acid leaching protocol includes a heating at 90°C during 10 min. This leaching protocol is issued from Berger et al. (2008) who recommend the heating to dissolve the refractory forms of metals bound with intracellular protein or intracellular trace metals in particles collected in river plumes, i.e. organic macromolecules bound trace metals (heating enabling the degradation of

these polymeric structures). Your arguments are based on the capacity of this protocol to reproduce acid digestion in gut krill.. Yet, the dissolution in this condition is probably more aggressive than gut krill due to this step of heating. The risk with this protocol is to access to refractory form which is never available for phytoplankton. Besides in your data, the solubility of Al is higher than the one of Fe with this protocol, isn't it due to leaching some aluminum from the refractory alumino-silicate minerals (see specific comments, P13, L400)? What is the relevance of this refractory forms of TEs for estimating their bioavailability? Do you have literature to support the fact of organo-metal complexes are bioavailable? In my opinion, it could be more interesting to use this 25% acetate soluble fraction as a proxy of organically bound trace metals and discuss on the link between source/instantaneous solubility and chemical form of trace metals, rather as an "upper limit" of solubility which is probably never reach in the natural conditions. . .

Specific comments :

The paper is a part of range of publications associated to Atlantic Geotraces campaigns. The presented data analysis is supported by numerous other works issued from these campaigns. The crucial information to understand the results is not always specified in the text and it's necessary to study the cited papers to understand the interpretation of results. For example, the back-trajectories are at the heart of the discussion in this paper, but they are not available in the text (see below). The categories of trace metals "lithogenic", "anthropogenically enriched" and "mixed source" which are issued from Shelley et al. (2015) appear P9, L336 without previous explanation. . .

Moreover, the structuration of the paper is not linear and implies several repetitions or meandering discussions, which drow up the conclusions. This paper has no clear guidance and is hard to follow. For example: In "Introduction": a first background on the leaching protocol between L67 and 86, then a second part of this background with repetition is presented between L95 and 124. Or In "Results and Discussion", the discussion on the ratio Fe and Al begins L255 up to L265 in session 3.1, then continues in L275, then is addressed in the next session 3.2 (L 295-335). The UHP water soluble

fraction starts to be discussed in the session 3.2., then be continued in the session 3.3. "Aerosol solubility". . . I advise to re-organise your paper, with a session: "metals origin" including a "clear" summary of Shelley et al. (2015) focused on their conclusions about origin of Fe and TEs as a function of provenance completed with a work on identification of sources from adapted bibliography then a second session presenting "solubility" not as a function of backtrajectories but a function of metals source (and in consequence with adapted figures).

Certain experimental points need also to be specified to clarify the conclusions of this work.

- P5, L148, please precise the cut-off diameter (or PM fraction: TSP, PM10..) of the aerosol sampler.

- P7, L226-L228 and session 3.4: The determination of nss-$SO_4^{2-}$ is calculated from soluble $Cl^-$. However, it's known that there is a potential depletion of $Cl^-$ during transport of sea salts due to the reactivity with anthropogenic gas in particular acid gas as $HNO_3$. The probability of this depletion increases with the increase of acidity (e.g. Kerminen et al., 1998; Yao et al., 2003; Newberg et al., 2005; Chi et al., 2015). In consequence, the fraction of nss-sulfate could be overestimated. Moreover, your method of calculation of acidity excluded all the organic acid, such as oxalic acid, formic acid. . .which are observed in marine atmosphere (e.g. Kawamura et al., 2017). Lastly, acidity measurement based exclusively on sulfate and nitrate implies that these species are in their acid forms. However, it's known that these compounds are associated to neutralizing compounds as $NH_4^+$ (Weber et al., 2016). In general, ammonium concentrations or $Cl$-depletion are used to estimate the aerosol acidity (Newberg et al., 2005; Hennigan et al., 2015). You mention all this literature and these works (L512-524) and you propose that "these approaches should be considered for future studies". Only even now, without at least an estimation of neutralised fraction, I think that your method suffers too much uncertainties to provide robust conclusion. So, the session 3.4 on the link between solubility and your "acidity" should be removed or completed with cation

measurements.

- P8, L238: The air masses of collected samples in the cruise GA01 are not shown. "High latitude dust" includes both dust from paraglacial regions and volcanic lands. Thus, even if the provenance is "high latitude dust", the origin could be different. For the longest distance between "high latitude" and R/V, a mixing with other sources could have occurred: the collected filters close to Europe (G1-G6) could be feed by various other sources (North of Europe, Europe, marine..) in comparison to the samples G12 or G14...Please show back-trajectories for at least one of the samples close to Europe and another close to North Atlantic.

- P8, L246: replace 2015 by 2014 in reference: Conway and John. And P20, L627 add 2014 in this reference.

- P8, L258: Even if the correlation between Fe and Al in "high latitude dust" samples (= Geovide samples, please be homogeneous between text and Figure 2) is good, the ratio Fe/Al is variable (Figure 3), suggesting various mineral source: please complete this discussion.

- P8, L260 to P9L265: "no correlation between Fe and Al in the samples of N. American (r2=0.153..) and marine (r2=0.016..) provenance". However in the figure 2b, the plots between Fe and Al concentrations show a R2= 0.983 for N. American aerosols, and R2=0.758 for Marine aerosol. Why is this difference between text and figures? On the basis of a bad correlation, you argue a strong influence of anthropogenic emissions on the Fe/Al ratio in N. American samples. Yet, the ratio (0.86) is consistent with a mineral origin. The quality of correlation changes all the discussion on the origin of Fe, so please clarify that.

- P9, L273 : You propose that "other sources are responsible for residual variance" for metals as Ni or Cd. It could be useful for the discussion on solubility to distinguish correlation (and consequent residual variance) between TEs and Al for each provenance as for Fe. The figure 3 seems show that all the TEs are correlated with Al in the N.

African samples, meaning probably that the anthropogenic influence is poor in these samples.

- P10, L322: "the most heterogeneous group", for what?

- P11, L336 : Why do you consider all the metals as anthropogenically enriched whatever the provenance whereas all these metals in dust samples (A1-A15) are probably originated from dust from the enrichment factor calculated in Shelley et al. (2015)?

- P11, L350 : Why do you use a PMF analysis with 2 profiles whereas at least 3 are potentially present in your samples : African dust, Anthropic and High latitude dust ? Did you take only trace metals in PMF anlaysis? Nitrate or nss-Sulfate are good proxies to discriminate anthropogenic source. Moreover, PMF is a stastistical method, even if the factor 2 is richest for "anthropogenic" metals, that doesn't mean that this factor doesn't include a mineral fingerprint, for example the high latitude dust. It's clear for the A1-A15 samples, the high contribution of finger 2 is probably due to the fact that metals from mineral fingerprint is taken into account in this factor. This method with your applied conditions is not sufficiently discriminating to be useful here. The discussion on PMF analysis should be removed, because it provides no appropriate information on the provenance of TEs.

- P12, L382-389: I don't understand why this paragraph is here? A bad cut and paste?

- P13, L398: Why do you present these data? It's not new that the solubility decreases with the total Al content (e.g. Baker et al., 1996) and your results don't provide new conclusion. On the contrary, it could be interested to plot the solubility of TEs (no Fe) as a function of atmospheric loading or Al concentrations. Firstly, that enables to valid or not your assumption on the role of scattering in our data in this trend. Secondly, it could be interested to compare the behaviour of Fe (a proxy of mineral dust) with the ones of anthropogenic trace metals.

- P13, L400: I wonder really if the highest Al solubility in comparison to Fe is not due

to the dissolution of refractory alumino-silicate minerals. Ti and Fe have the same behavior between UHP-water and 25% acetic acid dissolution in dust samples (figure 5) and both of them are in part as oxide in mineral dust, i.e. "refractory" even at high temperature. Please complete this discussion.

- P13, L403-410: This paragraph is unclear: the first sentence is too long and confused and what is the link between your data and the Madcow MODEL?

- P13, L411-420: This conclusion is interesting but without link with the results presented in this session.

- P14, L451: The discussion around these results is for me off topic. It is a pity that this paragraph be discussed only in terms of comparison between UHP water and seawater protocol, it would be interesting to discuss precisely in terms of origin of metals and solubility in the two protocols. But again, the assumption, which is that the origin of trace metals is homogeneous and anthropogenic, doesn't enable to identify a link between solubility and chemical form. For example, could you distinguish by coloring the different origin of metals in the figures 7 (as for Fe) to see if a same metal present different behaviours for solubility?

- Figures 5 and 6: No uncertainty is provided in your graphs for the data, could you add this information?

References:

Arnalds, O., Dagsson-Waldhauserova, P., and Olafsson, H.: The Icelandic volcanic aeolian environment: Processes and impacts :A review, Aeolian Research, 20, 176-195, https://doi.org/10.1016/j.aeolia.2016.01.004, 2016.

Baratoux, D., Mangold, N., Arnalds, O., Bardintzeff, J.-M., Platevoet, B., Grégorie, M., and Pinet, P.: Volcanic sands of Iceland – Diverse origins of aeolian sand deposits revealed at Dyngjusandur and Lambahraun, Earth Surf. Proc. Landforms, 36,1789–1808, 2011

Chi, J. et al. Sea salt aerosols as a reactive surface for inorganic and organic acidic gases in the Arctic troposphere. Atmos. Chem. Phys. 15, 11341–11353 (2015).

Formenti, P., Schütz, L., Balkanski, Y., Desboeufs, K., Ebert, M., Kandler, K., Petzold, A., Scheuvens, D., Weinbruch, S., and Zhang, D.: Recent progress in understanding physical and chemical properties of African and Asian mineral dust, Atmos. Chem. Phys., 11, 8231-8256, 10.5194/acp-11-8231-2011, 2011.

Formenti, P., Caquineau, S., Desboeufs, K., Klaver, A., Chevaillier, S., Journet, E., and Rajot, J. L.: Mapping the physico-chemical properties of mineral dust in western Africa: mineralogical composition, Atmos. Chem. Phys., 14, 10663-10686, 10.5194/acp-14-10663-2014, 2014.

Fuhlbrügge, S., Krüger, K., Quack, B., Atlas, E., Hepach, H., & Ziska, F. (2013). Impact of the marine atmospheric boundary layer conditions on VSLS abundances in the eastern tropical and subtropical North Atlantic Ocean. Atmospheric Chemistry and Physics, 13(13), 6345-6357. DOI: 10.5194/acp-13-6345-2013.

Furukawa, T. and Takahashi, Y.: Oxalate metal complexes in aerosol particles: implications for the hygroscopicity of oxalate-containing particles, Atmos. Chem. Phys., 11, 4289-4301, https://doi.org/10.5194/acp-11-4289-2011, 2011.

Hennigan, C. J., Izumi, J., Sullivan, A. P., Weber, R. J., and Nenes, A.: A critical evaluation of proxy methods used to estimate the acidity of atmospheric particles, Atmos. Chem. Phys., 15, 2775-2790, https://doi.org/10.5194/acp-15-2775-2015, 2015.

Kawamura, K., Hoque, M. M. M., Bates, T. S., and Quinn, P. K.: Molecular distributions and isotopic compositions of organic aerosols over the western North Atlantic: Dicarboxylic acids, related compounds, sugars, and secondary organic aerosol tracers, Organic Geochemistry, 113, 229-238, https://doi.org/10.1016/j.orggeochem.2017.08.007, 2017.

Kerminen V.-M., K. Teinilä, R. Hillamo, T. Pakkanen, Substitution of chloride in sea-salt

particles by inorganic and organic anions, Journal of Aerosol Science, 29, 929-942, 1998.

Lafon, S., I. N. Sokolik, J. L. Rajot, S. Caquineau, and A. Gaudichet, Characterization of iron oxides in mineral dust aerosols: Implications for light absorption, J. Geophys. Res., 111, D21207, doi:10.1029/2005JD007016, 2006.

Lázaro, F. J., Gutiérrez, L., Barrón, V., and Gelado, M. D.: The speciation of iron in desert dust collected in Gran Canaria (Canary Islands): Combined chemical, magnetic and optical analysis, Atmos. Environ., 42, 8987-8996, 2008.

Newberg, J. T., B. M. Matthew, and C. Anastasio (2005), Chloride and bromide depletions in sea-salt particles over the northeastern Pacific Ocean, J. Geophys. Res., 110, D06209, doi:10.1029/2004JD005446.

Óladóttir, B. A., Sigmarsson, O., Larsen, G., and Devidal, J-L.: Provenance of basaltic tephras from Vatnajökull subglacial volcanoes, Iceland as determined by major- and trace-element analyses, The Holocene, 21, 1037–1048, 2011.

Yao, X., Fang, M. & Chan, C. K. The size dependence of chloride depletion in fine and coarse sea-salt particles. Atmos. Environ. 37, 743–751 (2003).

Zhao, Y. & Gao, Y. Acidic species and chloride depletion in coarse aerosol particles in the US east coast. Sci. Total. Environ. 407, 541–547 (2008).
* * *

---

## Author Comment (AC1) · 8 Feb 2018

This manuscript describes the results obtained from different leaching methods used to liberate soluble trace elements from aerosol samples collected over the North Atlantic Ocean. The majority of the work focuses on leaches with ultra-high purity water (UHP) and an acetic acid solution, although some results obtained from leaching with seawater are also reported. The dataset presented is of high quality and it has been

subjected to a very thorough analysis. The manuscript is highly suitable for publication in Biogeosciences, although I feel that some clarifications of relatively minor points are necessary. Although the manuscript focuses on the UHP and acetic acid leach results, Section 3.3.3 presents results obtained by leaching a subset of the aerosol samples using sea- water. Currently the manuscript contains no information about how these experiments were performed. Please add this.

Thank you for your review. A description of this can be found starting at line 190, and Section 3.3.3. starts with, 'Seawater leaches were conducted on a subset of samples (GA03-2011), to investigate the suitability of seawater as the leach medium in the instantaneous leach.'

Non-seasalt (nss) sulfate concentrations in the aerosol samples were estimated using aerosol chloride concentrations as an indicator of seaspray content. I understand that this was the only indicator available, since major cations (e.g. sodium or magnesium ion concentrations) were not measured. However, chloride is not an inert species in marine aerosol, since it can be converted to hydrogen chloride and lost to the gas phase (Andreae and Crutzen, 1997). The extent of chloride loss is likely to be greater in polluted air types, such as those originating over Europe and North America, but all of the samples will be affected to some extent. Thus the nss concentrations presented here will be over-estimated and it would be helpful to note this in the text.

The description of the calculation of nss-$SO_4^{2-}$ has been removed as the aerosols acidity section, which used the nss-$SO_4^{2-}$ data, is no longer included in this manuscript.

I found it difficult to compare the results presented in the main body of the manuscript with those in the Supplementary Information (and in the two earlier manuscripts in which previous work on these samples was presented) because different labels have been used for the samples in different places. If it is not possible to only use a single set of labels, please could all the different labels be added to the tables in the supplement?

Thank you for drawing attention to this. This is a good point. In Shelley et al. (2017), which only discussed samples from GA01, the label 'A' referred to aerosol samples to differentiate them from the rain samples 1(R) in the figures. A note has been added to the caption to notify readers to this. A similar note was added to the caption for Table S1, as well as the labels being added in brackets after the GEOTRACES sample numbers. The labelling convention in Shelley et al. (2015, GA03 only), is the same as in this manuscript.

There is also a specific issue with the naming convention for sample M3-GA03. The "M" here (and the map in Shelley et al., 2015) imply that this was a Marine sample. The discussion on lines 328 -332 specifically state that it had a European air mass back trajectory. Please clarify this. You are correct. This was an error in the manuscript. In addition, further clarification and discussion has been added from Line 379.

I assume that positive matrix factorisation analysis (lines 350 onward) was done using total trace element concentrations. Please could this (or the correct information) be specifically stated. Correct. Added at Line 306.

All of Figures 2 – 7 would be improved by the addition of error bars. This would greatly assist the reader in putting the relatively high variability in calculated parameters (TE ratios to Al, or percentage soluble fractions) for samples with low total concentrations into context with the low variability, high total concentration samples with North African origin. We agree however, there are several reasons why error bars have not been added to the plots. The reason plots that include fractional solubility data on one or both axes don't have error bars is that replicate digests/leaches were not conducted for all of the samples, and with the exception of one sample, there are no replicates on the same sample for total and the soluble concentrations. Therefore, a SD for the fractional solubilities cannot be calculated. I appreciate that this is not ideal. Error bars could have been added to Fig. 2, but by doing so it is very hard to see the different symbols. For the elemental ratios, calculating the SD was not a problem, as it was possible to summed the SDs for the two elements being ratioed. However, the problem

is that by adding error bars to the plot it makes really hard to see the different shapes and colours of the symbols. As such, the plots have not got error bars added, but the SDs have been added in brackets after the relevant samples in Table S1. Where we are talking about a high degree of variability in the data, we are talking about within the aerosol source categories, rather than replicates of the same sample. This data has already plotted with error bars and can be found in the Supplementary Material, Figure S3. The data can be found in Tables S3 and S4.

Minor points: Line 95: "seawater" spelling. Corrected Lines 103-115. Much of this paragraph is repetition of material from previous paragraphs. It could easily be shortened. The introduction has been rewritten to reduce the repetition, and the sections referred to here now start at line 93. Line 319: The eruption of Eyjafjallajökull took place in 2010, not 2011. Corrected Line 353: I think Fig S2b should be referenced here, not S1b. Corrected Line 369: The panels of Fig. 4 are not labelled on the figure. This has been corrected. Line 374: "the ranges of fraction solubility" - I think there is a misspelling here. Corrected. Lines 382-387: This is a very long and cumbersome sentence. Please consider splitting it. This has been changed to, 'Furthermore, the ability of models to replicate subtleties in aerosol TE solubility may prove critical in forecasting ecosystem impacts and responses. Due to the magnitude of North African dust inputs to North Atlantic region, this is a particular challenge and is compounded by additional unknowns such as how aerosol acidity will be impacted by the combined effects of increasing industrialisation/urbanisation, and changes in the magnitude of future mineral dust supply and biomass burning (Knippertz et al., 2015; Weber et al., 2016).' Starting at line 553. Lines 444-445: "The differences source dependence of" Please correct. Corrected Line 481: Panel h of Fig 6 shows data for cadmium. Lead data is on panel i. Corrected Lines 509-511: This statement is very speculative. Please add further explanation or consider removing. The section on aerosol acidity has been removed. Line 534: I am a little confused by the opening statement of this sentence. This manuscript has been devoted to the direct measurement of TE solubility! This sentence has been removed. The original point was that we don't measure solubility directly, but calculate

it from leach data that is sensitive to differences between the various leach protocols. Lines 536-537: "in regions of high mineral dust deposition and/or productivity fractional solubility". An odd construction. What is productivity fractional solubility? Corrected – this was missing a comma between productivity and fractional solubility

Lines 538-839: How is it possible to have an inverse relationship between TE fractional solubility and aerosol provenance? Corrected – provenance has been removed

---

## Author Comment (AC2) · 8 Feb 2018

This paper reports Fe, Al and trace metals composition and solubility in aerosols particles collected in North Atlantic Ocean. Even if it exists a lot of data on the solubility of Fe and major trace metals in Atlantic area (for ex. Clivar, AMT. campaigns), new data and in particular new perspective for using this data is always interesting. In this idea, the title of this manuscript was promising but finally, it is rather disappointing after

reading. Indeed, even if the title implies a discussion on the relation between solubility and aerosol origin, the paper is limited to a dataset of solubility values as a function of air mass back-trajectories. It's very frustrating because the material is present to make this study original and significant. Another purpose of paper is the investigation to different leaching protocol for solubility measurement standardization. This twofold topic (provenance and protocol) makes abstruse the main objective of the paper. Thank you for your review. The discussions about standardisation have been removed.

Although this experimental work has been carefully conducted and contains interesting results, it is short of new findings in this current state due to a lack of guidance in the discussion on the results and of comparison with the literature. Consequently, I strongly encourage the authors to work again on this manuscript because your data could bring appreciable and consistent results for the community. The major highlights proposed in this paper are "trace elements from aerosols from 1) North Africa were always the least soluble, and the most homogeneous . . ., 2) aerosols from the most remote locations were generally the most soluble, but had the most spread in the values of fractional solubility and 3) primarily pollution-derived TEs (Ni, Cu, Zn, Cd and Pb) were significantly enriched above crustal values in aerosols, even in samples of North African origin." The focus has changed: the main findings are now that (1) there are exceptions to the general trend that fractional solubility of TEs is inversely related to atmospheric loading. The fractional solubility of Mn, Zn and Cd appears to be independent to atmospheric loading, and (2) air mass back trajectories are not sufficiently discriminating to identify aerosol source.

The critical point on this paper is the signification of term "provenance". In the manuscript, the aerosol provenance is considered as the "back-trajectory". Firstly, the back-trajectories were made for an arrival height of 500m, whereas the maximum altitude of boundary layer is often between 200 and 600 m in North Atlantic (Petenko et al., 1996; Fuhlbrügge et al., 2013), and aerosol sampling was in the boundary layer, so it is not automatically consistent. The GA01 back trajectories presented in the Supplementary Material of Shelley et al. (2017) were simulated for arrival heights of 50, 500 and 1500 m. The back trajectories for GA03, presented in Shelley et al. (2015) have been redrawn with the same three arrival heights as the GA01 samples (50, 500 and 1500 m), to include at least one arrival height in the MBL. The four representative AMBTs from GA03 and all AMBTs from GA01 can now be found in the Supplementary Material (Fig. S1).

Moreover, the "loading" in various aerosols is not only dependent on the trajectory but also on emissions along the transport of air masses. A same air masses can be a mix between various types of aerosols. In consequent, as solubility is dependent on kind of particles, it's obvious to observe a largest variability of solubility for aerosols from most remote locations in comparison to dust samples (findings 1) and 2)). In one case, it's a same source and in the other case that includes various sources and hence kind of particles. The relevant conclusion of this work should precisely be that aerosol provenance is not sufficient to estimate composition and solubility of trace metals. This is now a key conclusion to the paper. However, the regional groupings, as determined by AMBT, are retained as a way of grouping the data to look at regional variations in fractional solubility.

From your database and the previous conclusions of Shelley et al. (2015) on the identification of sources of TEs (not provenance) in aerosols, further investigations could enable to emphasize a relationship between TEs solubility and origin. In this purpose, the origin of aerosol is a huge question which demands a more extensive bibliography to provide convincing proof of metals sources. A part of this comparison with bibliography is provided in the session 3.1 and 3.2. Nonetheless, the structuration of the paper is to much confuse to extract the pertinent information (see specific comments). These sessions should only include a summary of appropriate results from Shelley et al. (2015) for the discussion on the link with solubility. Moreover, a relevant bibliography is often missing in the manuscript. Several field campaigns during the last decade (AMMA, DODO, DABEX or SAMUM) + specific works (e.g. Trapp et al., 2010) permitted to improve our knowledge about the African dust composition before and after transport. Even if this literature is focused on major metals as Al, Fe or Ti, this literature about Saharan dust characterisation should be used to validate/discuss your aerosol origin. Moreover, the category "high latitude dust" is very extended with Al/Fe ratio ranging from 0.1 to 1 (L322-334), suggesting a variability of aerosol sources in these samples (see specific comments). The conclusion of authors about the mixed volcanic and anthropogenic origins of GA01 samples is supported only by the back-trajectories (not shown) and a comparison with the TEs concentrations in volcanic ash from Achterberg et al. (2013), but no comparison on elemental ratios is provided. Is this variability is consistent with the typical composition of volcanic or cold environment dust? Again, several papers report Fe or metals content in high latitude volcanic regions producing dust as Iceland (e.g. Baratoux et al., 2011, Óladóttir et al., 2008 and 2011). It could be interesting to compare with these data for discriminating the origin of this high latitude dust. Further discussion of the high latitude dust sources is now included in the text, and a table of elemental ratios from the various studies in regions that contribute aerosols to the North Atlantic is included in the Supplementary Material (Table S2).

The third conclusion of a paper is that polluted derived TEs were significantly enriched above crustal values in aerosol and notably in dust samples (A1-15), on the basis of comparison with UCC. This conclusion is already in Shelley et al. (2015) for GA03 cruises. The new information could be for GA01 for high latitude dust, but this work is not done. Furthermore, I'm not agree with your conclusion on the enrichment "even in samples of North African origin". As a matter of facts, Shelley et al. (2015) conclude that except Cd, the "pollution-derived" elements present EFs less than 10 with respect to Al in the African samples (See Shelley et al., 2015, figure 6). Moreover, your main argument is that these elements have some significant increase from the UCC mass ratio in your dust samples (see specific comments: P11,L336). However, it's known that the production of mineral dust is associated to a chemical fractioning due to the size partitioning between coarse rich-Si grains and the finest clay fraction during sandblasting. It means an enrichment in Al, Fe, Ca. in dust particles in comparison to soil

(Lafon et al., 2006). Thus, the authors are surprised at Al/Fe ratio around 0.76 (P9, L275). Yet, this value is fully consistent with the common knowledge on African dust (see previous comment on bibliography + e.g. Formenti et al., 2011 or 2014 and ref therein or Lazaro et al., 2008). Due to the trace metals partitioning in soil size fraction, this fractioning and hence enrichment probably happens also for trace metals in mineral dust. Your data seem to confirm this trend, previously observed by Trapp et al. (2010) after long-range transport. it's bad that this point of view is not addressed here. . . The original intention was to argue that the UCC ratio is not representative of North African dust inputs, but that the ratio we observed is consistent with other studies. We don't think that the North African dust is enriched with pollution-derived elements, but that there could be a component of aerosols coming from Europe mixed in with the Saharan end-member, which is also consistent with other studies (e.g. Baker and Jickells, 2017). A new table has been included in the Supplementary Material (Table S1), which includes elemental ratios from studies in the AMBT regions. It does not include literature data from a Saharan end-member as this was discussed extensively in Shelley et al. (2015).

Finally, a part of the manuscript is focused on the standardization of methods to estimate TE solubility. The conclusion on this session is not convincing for me because the proposed "upper limit" is not supported by the "reality" of aerosol dissolution in atmospheric or sea waters. The 25% acetic acid leaching protocol includes a heating at 90âŮȩC during 10 min. This leaching protocol is issued from Berger et al. (2008) who recommend the heating to dissolve the refractory forms of metals bound with intracellular protein or intracellular trace metals in particles collected in river plumes, i.e. organic macromolecules bound trace metals (heating enabling the degradation of these polymeric structures). Your arguments are based on the capacity of this protocol to reproduce acid digestion in gut krill. Yet, the dissolution in this condition is probably more aggressive than gut krill due to this step of heating. The risk with this protocol is to access to refractory form which is never available for phytoplankton. Besides in your data, the solubility of Al is higher than the one of Fe with this protocol, isn't it due

to leaching some aluminum from the refractory alumino-silicate minerals (see specific comments, P13, L400)? What is the relevance of this refractory forms of TEs for estimating their bioavailability? Do you have literature to support the fact of organo-metal complexs are bioavailable? In my opinion, it could be more interesting to use this 25% acetate soluble fraction as a proxy of organically bound trace metals and discuss on the link between source/instantaneous solubility and chemical form of trace metals, rather as an "upper limit" of solubility which is probably never reach in the natural conditions.

We absolutely agree that this is an upper limit, and potentially an over-estimation of the upper limit. However, we feel that the use of the heating step is justified as we wanted to use exactly the same protocol as the SPM leaches done on GA01, and some GA03 samples to allow direct comparison between the two datasets. A key goal of this work was to link the atmospheric inputs to processes occurring in the ocean. Data from the aerosol leaches and dissolved and particulate trace elements indicate that atmospheric inputs are not the dominant source of Al, Fe or Pb along the GA01 transect (Menzel-Barraqueta et al.,. Tonnard et al., and Zurbrick et al., submitted to this special issue).-However, we have acknowledged that there could be an overestimation of the upper-limit of solubility in the text (from line 208). We feel that if refractory metal is liberated it has the potential to be bioavailable to some micro-organisms, e.g. Tricho (Rubin et al., 2011). In future, it would be a good idea to conduct experiments to test the difference in fractional solubility estimates using this protocol with/without the heating step on different types of aerosols.

In answer to your question about the bioavail ability of organically-complexed TEs, the review by Shaked and Lis (2012) investigates this question, and provides examples of literature that supports this. for Fe. This paper is cited in the manuscript.

Specific comments : The paper is a part of range of publications associated to Atlantic Geotraces campaigns. The presented data analysis is supported by numerous other works issued from these campaigns. The crucial information to understand the results is not always specified in the text and it's necessary to study the cited papers to understand the interpretation of results. For example, the back-trajectories are at the heart of the discussion in this paper, but they are not available in the text (see below). The categories of trace metals "lithogenic", "anthropogenically enriched" and "mixed source" which are issued from Shelley et al. (2015) appear P9, L336 without previous explanation. . . Additional wording has been added to this text that we hope provides suitable clarification. The text starting at Line 454 now reads "Figure 5 highlights the distinction between the lithogenic elements, Al, Fe and Ti (universally low solubility in UHP water, mostly < 20 %, and extremely low solubility of North African aerosols, < 1 %), and the anthropogenic, pollution-dominated elements, Ni, Cu, Zn, Cd and Pb (solubility up to 100 %). Manganese (Mn) and Co have both lithogenic and anthropogenic sources, so are classified as "mixed-source", and have intermediate solubilities."

Moreover, the structuration of the paper is not linear and implies several repetitions or meandering discussions, which drow up the conclusions. This paper has no clear guidance and is hard to follow. For example: In "Introduction": a first background on the leaching protocol between L67 and 86, then a second part of this background with repetition is presented between L95 and 124. Or In "Results and Discussion", the discussion on the ratio Fe and Al begins L255 up to L265 in session 3.1, then continues in L275, then is addressed in the next session 3.2 (L 295-335). The UHP water soluble fraction starts to be discussed in the session 3.2., then be continued in the session 3.3. "Aerosol solubility". I advise to re-organise your paper, with a session: "metals origin" including a "clear" summary of Shelley et al. (2015) focused on their conclusions about origin of Fe and TEs as a function of provenance completed with a work on identification of sources from adapted bibliography then a second session presenting "solubility" not as a function of back trajectories but a function of metals source (and in consequence with adapted figures). The problem with repetition has been addressed through restructuring and rewriting sections. There are new sections headings: 3.3.1. Identifying sources of TEs. The 'solubility' section is still structured by AMBT category because clear sources could not be identified that would have enabled reorganising the data accordingly. However, a new section has been included (Section

3.4) at the end of the manuscript that discusses sub-groups within the AMBT categories that are suggested by cluster analysis of the total TE data and the fractional solubility of TEs.

Certain experimental points need also to be specified to clarify the conclusions of this work. - P5, L148, please precise the cut-off diameter (or PM fraction: TSP, PM10.) of the aerosol sampler. TSP has been added to text. Line 138.

P7, L226-L228 and session 3.4: The determination of nss-SO42- is calculated from soluble Cl-. However, it's known that there is a potential depletion of Cl- during transport of sea salts due to the reactivity with anthropogenic gas in particular acid gas as HNO3. The probability of this depletion increases with the increase of acidity (e.g. Kerminen et al., 1998; Yao et al., 2003; Newberg et al., 2005; Chi et al., 2015). In consequence, the fraction of nss-sulfate could be overestimated. Moreover, your method of calculation of acidity excluded all the organic acid, such as oxalic acid, formic acid. . .which are observed in marine atmosphere (e.g. Kawamura et al., 2017). Lastly, acidity measurement based exclusively on sulfate and nitrate implies that these species are in their acid forms. However, it's known that these compounds are associated to neutralizing compounds as NH4+ (Weber et al., 2016). In general, ammonium concentrations or Cl-depletion are used to estimate the aerosol acidity (Newberg et al., 2005; Hennigan et al., 2015). You mention all this literature and these works (L512-524) and you propose that "these approaches should be considered for future studies". Only even now, without at least an estimation of neutralised fraction, I think that your method suffers too much uncertainties to provide robust conclusion. So, the session 3.4 on the link between solubility and your "acidity" should be removed or completed with cation measurements. As suggested, this section has been removed.

P8, L238: The air masses of collected samples in the cruise GA01 are not shown. "High latitude dust" includes both dust from paraglacial regions and volcanic lands. Thus, even if the provenance is "high latitude dust", the origin could be different. For the longest distance between "high latitude" and R/V, a mixing with other sources could

have occurred: the collected filters close to Europe (G1-G6) could be feed by various other sources (North of Europe, Europe, marine) in comparison to the samples G12 or G14...Please show back-trajectories for at least one of the samples close to Europe and another close to North Atlantic. Agreed. All of the AMBTs from GA01 have been reproduced and can be found as Figure S1 in the Supplementary Material. The four representative AMBTs from GA03 have been redrawn and are also included in Figure S1.

P8, L246: replace 2015 by 2014 in reference: Conway and John. And P20, L627 add 2014 in this reference. Done.

P8, L258: Even if the correlation between Fe and Al in "high latitude dust" samples (= Geovide samples, please be homogeneous between text and Figure 2) is good, the ratio Fe/Al is variable (Figure 3), suggesting various mineral source: please complete this discussion. This discussion can now be found starting at line 345.

P8, L260 to P9L265: "no correlation between Fe and Al in the samples of N. American (r2=0.153..) and marine (r2=0.016..) provenance". However in the figure 2b, the plots between Fe and Al concentrations show a R2= 0.983 for N. American aerosols, and R2=0.758 for Marine aerosol. Why is this difference between text and figures? On the basis of a bad correlation, you argue a strong influence of anthropogenic emissions on the Fe/Al ratio in N. American samples. Yet, the ratio (0.86) is consistent with a mineral origin. The quality of correlation changes all the discussion on the origin of Fe, so please clarify that. This was a mistake and has been corrected, starting at line 285.

P9, L273 : You propose that "other sources are responsible for residual variance" for metals as Ni or Cd. It could be useful for the discussion on solubility to distinguish correlation (and consequent residual variance) between TEs and Al for each provenance as for Fe. The figure 3 seems show that all the TEs are correlated with Al in the N. African samples, meaning probably that the anthropogenic influence is poor in these samples. This is not done as a new Figure 4 and discussion has been added instead,

which includes all TEs under discussion.

P10, L322: "the most heterogeneous group", for what? Of the Fe/Al ratios. This wording has been deleted. It now reads, 'In contrast, samples from the most remote locations, the Marine and High Latitude aerosols, had the most spread in their fractional solubility and elemental ratios....'. Starting at line 604.

P11, L336 : Why do you consider all the metals as anthropogenically enriched whatever the provenance whereas all these metals in dust samples (A1-A15) are probably originated from dust from the enrichment factor calculated in Shelley et al. (2015)? We don't, we had previously argued that the samples had a relatively low concentration of Al compared to other elements.

P11, L350 : Why do you use a PMF analysis with 2 profiles whereas at least 3 are potentially present in your samples : African dust, Anthropic and High latitude dust ? Did you take only trace metals in PMF anlaysis? Nitrate or nss-Sulfate are good proxies to discriminate anthropogenic source. Moreover, PMF is a stastistical method, even if the factor 2 is richest for "anthropogenic" metals, that doesn't mean that this factor doesn't include a mineral fingerprint, for example the high latitude dust. It's clear for the A1-A15 samples, the high contribution of finger 2 is probably due to the fact that metals from mineral fingerprint is taken into account in this factor. This method with your applied conditions is not sufficiently discriminating to be useful here. The discussion on PMF analysis should be removed, because it provides no appropriate information on the provenance of TEs. The model is not stable with more than two factors, due to the relatively small dataset. This is why only two factors were used. We have tried to remove the North African samples to look for other groups of TEs that would be diagnostic of other sources, but a crustal factor is always one of the two factors. We also tried this with excess metal (assuming the N. African ratio as the reference ratio), but the same thing happened. We also tried adding the NO3- data to the analysis, but with no new information generated. We agree that the PMF provides limited evidence for sources other than mineral dust, but include this figure (Fig. S2)

in the discussion because of its inability to identify sources, and as a reason to try another multivariate statistical approach (cluster analysis) which is presented later in the manuscript.

P12, L382-389: I don't understand why this paragraph is here? A bad cut and paste? This sentence has been reworded, "Furthermore, the ability of models to replicate subtleties in aerosol TE solubility may prove critical in forecasting ecosystem impacts and responses. Due to the magnitude of North African dust inputs to North Atlantic region, this is a particular challenge and is compounded by additional unknowns such as how aerosol acidity will be impacted by the combined effects of increasing industrialisation/urbanisation, and changes in the magnitude of future mineral dust supply and biomass burning (Knippertz et al., 2015; Weber et al., 2016)." Starting at line 553.

P13, L398: Why do you present these data? It's not new that the solubility decreases with the total Al content (e.g. Baker et al., 1996) and your results don't provide new conclusion. On the contrary, it could be interested to plot the solubility of TEs (no Fe) as a function of atmospheric loading or Al concentrations. Firstly, that enables to valid or not your assumption on the role of scattering in our data in this trend. Secondly, it could be interested to compare the behaviour of Fe (a proxy of mineral dust) with the ones of anthropogenic trace metals. Figure 4 has been redrawn and replaced using all TEs. The main conclusion is that Mn, Zn and Cd do not follow the same trend of having an inverse relationship with atmospheric loading as Fe and Al.

- P13, L400: I wonder really if the highest Al solubility in comparison to Fe is not due to the dissolution of refractory alumino-silicate minerals. Ti and Fe have the same behavior between UHP-water and 25% acetic acid dissolution in dust samples (figure 5) and both of them are in part as oxide in mineral dust, i.e. "refractory" even at high temperature. Please complete this discussion. This is done starting at line 432. 'Although, we should not rule out that this effect is the result of the heating step in the 25 % acetic acid leach attacking the alumino-silicate matrix. Further experimentation with and without the heating step should resolve this issue.'

[Figure]

P13, L403-410: This paragraph is unclear: the first sentence is too long and confused and what is the link between your data and the Madcow MODEL? This has been moved to a new section (Section 3.5) as we wanted to draw attention to how important it is to accurately parameterise fractional solubility. This section also includes a short discussion on modelling Fe and Mn.

P13, L411-420: This conclusion is interesting but without link with the results presented in this session. This conclusion is now supported with evidence from this study and starts at line 570.

P14, L451: The discussion around these results is for me off topic. It is a pity that this paragraph be discussed only in terms of comparison between UHP water and seawater protocol, it would be interesting to discuss precisely in terms of origin of metals and solubility in the two protocols. But again, the assumption, which is that the origin of trace metals is homogeneous and anthropogenic, doesn't enable to identify a link between solubility and chemical form. For example, could you distinguish by coloring the different origin of metals in the figures 7 (as for Fe) to see if a same metal present different behaviours for solubility? Figure 6 – the samples have been colour-coded and a table (Table 1) has been added that shows which groups of samples have slopes that do not differe significantly from 1.0. We do not think that the origin of the metals are homogeneous (although the North African sample TE composition is more homogeneous that for the other groups), for the precise reason that multiple sources contribute to the aerosols in each region, as well as mixing occurring en route. Perhaps this is why no obvious relationship is seen between the samples being leached with either UHP water or seawater. Also, perhaps primarily because of the amount of mixing before collection, so far from sources, we are having difficulty identifying sources.

Figures 5 and 6: No uncertainty is provided in your graphs for the data, could you add this information? Error bars have not been added because of the problem with determining the SD of the fractional solubility. In Figure 5, the addition of error bars makes it impossible to distinguish the different symbols.

Please also note the supplement to this comment:
https://www.biogeosciences-discuss.net/bg-2017-415/bg-2017-415-AC2-supplement.pdf

---

## Author Comment (AC4) · 8 Feb 2018

[revised manuscript text omitted]

For most of the other TEs investigated here (Ti, Co, Ni, Cu, Zn, Cd and Pb; Fig. 4a), the same general
trend as Fe and Al was observed following the UHP water leach. For the 25 % acetic acid leaches
(Fig. 4b), Ti, Ni, Cu and Pb solubility decreased with an increase in atmospheric loading, but the trend
was less clear for Co. For Co the inverse relationship between UHP water solubility and loading was
not observed when using the 25 % acetic acid leach, most likely because Co may be associated with
the Mn and Fe oxides that are easily reduced using this leach. In contrast, Mn, Zn and Cd did not
display the same inverse relationship as Fe and Al after either leaching step (Figs. 4a and b). This has
previously been noted for Mn (Jickells et al., 2016). Although the average fractional solubilities of Zn
and Cd (Zn: $37 \pm 28$ % and $55 \pm 30$ %, Cd: $39 \pm 23$ % and $58 \pm 26$ % for ultra-high purity water and
25 % acetic acid leaches, respectively) were similar to Mn ($32 \pm 13$ % and $49 \pm 13$ % for ultra-high
purity water and 25 % acetic acid leaches, respectively), the range was greater, with several samples
from different regions (although not North Africa) being 100% soluble after the second leach.

**3.3.2. Solubility of TEs: UHP water (instantaneous) compared to 25 % acetic acid leaches**
All ten TEs from the five different provenances were less soluble in UHP water than 25 % acetic acid
(Fig. 5). This is not a surprising finding given the lower pH of acetic acid compared with UHP water,
that acetate is a bidentate ligand, and the longer contact time of the aerosols with the leach solution in
the 25 % acetic acid leach procedure. In addition, there is some degree of source-dependent variability
in the relative proportions of each TE that is released by the two leaches. In general, as with the leaches with UHP water, the North African aerosols were distinctly less soluble than aerosols from the other source regions (Fig. 5). Figure 5 highlights the distinction between the lithogenic elements, Al,

Fe and Ti (universally low solubility in UHP water, mostly < 20 %, and extremely low solubility of

North African aerosols, < 1 %), and the anthropogenic, pollution-dominated elements, Ni, Cu, Zn, Cd and Pb (solubility up to 100 %). Manganese (Mn) and Co have both lithogenic and anthropogenic sources, so are classified as "mixed-source", and have intermediate solubilities. Like all the TEs reported here, Mn solubility in UHP water was significantly less (p < 0.01, two-tailed, homoscedastic t-test) in North African aerosols (median solubility = 19 %) than in the non-North African samples (median = 38%), which seems to contrast somewhat with the findings of Baker et al. (2006b) and

Jickells et al. (2016). However, in common with these earlier studies (Baker et al., 2006b; Jickells et al., 2016), there was no significant source-dependent difference in Mn solubility in 25 % acetic acid (non-North African samples: 49 ± 15%, North African samples: 49 ± 6.4%).

**3.3.3. Soluble TEs: UHP water compared to seawater instantaneous leaches**

Seawater leaches were conducted on a subset of samples (GA03-2011), to investigate the suitability of seawater as the leach medium in the instantaneous leach (Fig. 6). During this study, Fe solubility in seawater was lower than in UHP water (Fig. 6c). This phenomenon has previously been observed in atmospheric aerosols from the North Atlantic Ocean (Buck et al., 2010).  For Fe, only a few samples of North American and Marine provenance conformed to the relationship described by the equation proposed by Buck et al. (2010), with most of our data plotting above the regression line of the Buck et al. (2010) study (Fig. 6c), indicating that our data was relatively more soluble in UHP water compared to seawater than in this earlier study. One possibility is that the higher aerosol Fe loadings we observed during GA03-2011 (this study, maximum = 5650 ng Fe m$^{-3}$), compared to the A16N-2003

transect (Buck et al. 2010; maximum =1330 ng Fe m$^{-3}$), resulted in a particle concentration effect (Baker and Jickells, 2006), whereby the relationship between aerosol Fe loading and fractional solubility breaks down because  dust on the filter can be a source of soluble Fe but can also scavenge dissolved Fe from the sea water leach solution as it passes through the filter.  Given that the link between Fe solubility in seawater and Fe-binding ligand availability is well established (e.g. Rue and

Bruland, 1995; Gledhill and Buck, 2012), an alternative explanation for the difference in Fe solubility is that the organic composition of the seawater used as the leach mediums differed between the two studies.

Mn is the only TE that has a slope close to unity (0.98; Fig. 6b), suggesting that solubility estimates were not impacted by the choice of leach medium used. This is consistent with other studies that have found that Mn solubility is less sensitive to the choice of leach media, or to aerosol provenance than other TEs (Baker et al., 2006b; Jickells et al., 2016). Due to the large variability in the data set, there was no significant difference between Mn solubility in UHP water or seawater (32 ± 13 % and 24 ± 17

%, respectively; Fig. S3, and Tables S3 and S4, Supplementary Material). Table 1 shows which regions had slopes for UHP water versus seawater fractional solubility that did not differ significantly from 1.0 at the 95 % confidence level (t-statistic; Table S5). For the North American samples, the slope did not differ significantly from 1.0 for Al, Mn, Co, and Zn. For the Marine samples, the same was true for Al, Co and Zn, and for the North African samples this was the case for Al, Fe, Ni, Cu, Zn and Cd. Pb was the only TE with slopes differing significantly from 1.0 for all regions.

[revised manuscript text omitted]

**Captions: Figures**

Figure 1. The GEOTRACES GA01 and GA03 cruise tracks (GA01, GA03-2010 and GA03-2011). In total, 57 aerosol samples (GA01 n = 18, GA03 n = 39; black dots) were collected. The samples are grouped by aerosol source region (green = European (E1-4), blue = Marine (M1-12), yellow = North African (A1-15), red = North American (N1-8), and grey = High Latitude (G1-18)), identified from air mass back trajectory simulations using the NOAA ARL model, HYSPLIT (Stein et al., 2015; Rolph, 2017). Note that a different labelling convention was used in Shelley et al. (2017) to refer to the GA01 samples. Here we use G1-18 to refer to the samples collected during GA01 (A1-18 in Shelley et al., 2017), and A1-15 to refer to the North African samples from GA03.

Figure 2. Total aerosol Fe and Al (ng m$^{-3}$) for: (a) all aerosol samples from cruises GA01 and GA03, (b) samples from sources other than North Africa (i.e. the black diamonds in Fig. 2a), and (c) the samples inside the dashed box in Fig. 2b. For High Latitude dust n = 18, European samples n = 4, North American samples n = 8, Marine samples n = 12, and Saharan samples n = 15. Note error bars (standard deviations shown in Table S1) are not included so as not to obscure the symbols.

Figure 3. Elemental mass ratios (normalised to Al) of total (black circles) and UHP water soluble (white triangles) TEs. The UCC elemental ratio (Rudnick and Gao, 2003) is indicated by the solid horizontal line, and the elemental ratio in North African sourced aerosols (Shelley et al., 2015) is indicated by the dashed horizontal line on each plot. The red vertical lines separate the aerosol source regions, which are labelled in panel (a). Samples G9-GA01 and M3-GA03 are indicated by blue arrows in panel c (see text for details).

Figure 4. (a) Percentage of UHP water soluble TEs versus total aerosol TE (ng m$^{-3}$), (b) percentage of 25 % acetic acid soluble TE versus total aerosol TE (ng m$^{-3}$). Data is plotted on log-log scales.

Figure 5. Solubility of Al, Ti, Mn, Fe, Co, Ni, Cu, Zn, Cd, Pb following a UHP water leach (UHP water, black circles, calculated using Eq. 1), and a sequential leach of 25 % acetic acid (HAc, grey squares, calculated using Eq. 2). The red vertical dashed lines represent the different aerosol source categories, as labelled in panel (b). Note that Ti (panel b) is highly insoluble and has a maximum value of <13%.

Figure 6. Comparison of TE solubility following instantaneous leaches using UHP water or locally-collected, filtered seawater. The solid line is the 1:1 line. Where fewer data are observed, concentrations were below detection for one or both of the two leaches. The data for soluble aerosol Fe from within our study region from Buck et al. (2010) are plotted as black triangles in panel (c).

Figure 7. Heirachical cluster analysis of (a) log transformed total TE concentration data plus NO3-, and (b) log transformed fractional solubility following the two-step sequential leach (fractional solubility calculated using Eq. 2). The coloured blocks correspond with the aerosol source regions shown in the legend.

**Captions: Tables**

Table 1. Slopes that did not differ significantly from 1.0 at the 95 % confidence levels for the UHP water versus seawater instantaneous leaches are marked with x. NA = not assessed due to the number of paired samples being ≤ 3.

---

## Author Comment (AC5) · 8 Feb 2018

Revised Supplementary Material attached

Please also note the supplement to this comment:
https://www.biogeosciences-discuss.net/bg-2017-415/bg-2017-415-AC5-supplement.pdf

---

## Author Response (AR2)

**Response to the editor**

**Many thanks for your suggestions for improvement to our manuscript. Our responses are detailed below and are followed by a version of the submitted manuscript showing all mark ups.**

l. 434: 'Although we...'

**Corrected**

l. 515 and caption Fig. 7: 'Hierarchical' instead of 'Heirachical'

**Corrected**

l. 535: 'This suggests that...'

**Corrected**

l. 543: 'The samples found near Greenland...'

**Changed to, "The samples collected near Greenland…"**

Figure 6: the symbols for N. America and N. Africa appear in the version on the site almost with the same color (and same shape, which does not help)

**The same colours have been used for consistency with the other figures used in this manuscript, but the plots have been redrawn with different symbols to make it easier to differentiate between the N. American samples (red squares) and the N. African samples (orange diamonds).**

Figure 7: The caption for North Africa appears as orange-colored, whereas the color on the plot seems to be 'yellow'...

**The colour scheme has been maintained but a note in the caption has been added that notifies readers that the N. African samples are indicated by the yellow blocks of colour.**

[revised manuscript text omitted]